# Dust tides and rapid meridional motions in the Martian atmosphere during major dust storms

Zhaopeng Wu [1,2,3]*, Tao Li [3,4]*, Xi Zhang[5], Jing Li [1] & Jun Cui[1,2,3]

The atmosphere of Mars is strongly affected by the spatial and temporal variability of airborne dust. However, global dust variability within a sol (Martian day) is still poorly understood. Although short-term dynamic processes are crucial, detailed comparisons of simulated diurnal variations are limited by relatively sparse observations. Here, we report the discovery of ubiquitous, strong diurnal tides of dust in the Southern Hemisphere of Mars. Driven by the westward-propagating migrating diurnal thermal tide, zonally distributed dust fronts slosh back and forth in a wide latitudinal range of up to 40° within one sol during major dust storms. Dust tides—tidal transport of dust in this way—rapidly transport heat and constituents meridionally, allowing moist air near the summer pole to be rapidly transported to lower latitudes during the night, where it then can be lifted by daytime deep convection and contribute to hydrogen escape from Mars during global dust storms.

[1] School of Atmospheric Sciences, Sun Yat-sen University, Zhuhai, Guangdong 519082, PR China. [2] CAS Key Laboratory of Lunar and Deep Space Exploration, National Astronomical Observatories, Chinese Academy of Sciences, Beijing 100012, PR China. [3] CAS Center for Excellence in Comparative Planetology, Hefei, Anhui 230026, PR China. [4] CAS Key Laboratory of Geospace Environment, School of Earth and Space Sciences, University of Science and Technology of China, Hefei, Anhui 230026, PR China. [5] Department of Earth and Planetary Sciences, University of California Santa Cruz, Santa Cruz, CA 95064, USA. *email: wuzhp9@mail.sysu.edu.cn; litao@ustc.edu.cn

The spatial and temporal variability in airborne dust plays an essential role in the Martian climate system via its influence on the atmospheric thermal and dynamical states[1–3]. Observations have shown interannual, interseasonal, and diurnal variability in the dust distributions[4–8]. The rapid vertical transport of dust and water to high altitudes in the mid-to-high latitudes within just a few sols during global dust storms was recently discovered[9,10]. This short-term dynamic process on dusty Mars is intriguing. However, to date, a comprehensive characterization of the diurnal variation in global dust has not been conducted at a high-temporal resolution[11,12]. Most spacecraft observations are obtained from sun-synchronous orbit, typically viewing at only two day-and-night local times per sol[1].

Based on those observations with limited local-time coverage, high-altitude dust in the tropical atmosphere on Mars is found to exhibit large day–night variations[13–15]. However, the underlying mechanism is elusive. Previous works attributed it to several possible factors: pseudo-moist convection, orographic updrafts, scavenging by water ice, tidal processes, etc.[13–16]. It is difficult to disentangle these complicated and mixed factors. Fortunately, extreme weather events on Mars, such as major dust storms, can be treated as natural control experiments that may intensify one factor over the others. For instance, the extratropical diurnal tide —global oscillation in the atmosphere with a period of a solar day[17]—is dramatically enhanced in the southern hemisphere during major dust storms[18,19]. Numerical simulation studies also suggested the possible role of the thermal tide in modulating the mid-to-high-latitude dust distribution[15,16].

Unlike the gravity-driven ocean tide on Earth, atmospheric tides on Mars are thermally driven through the absorption of sunlight by the widely distributed dust and water ice in the atmosphere[3,18,20–23]. Once excited, thermal tides can propagate zonally and vertically and transport energy and momentum away from the excitation sources to other atmospheric layers[17,24,25]. One of the dominant thermal tides found in the Martian atmosphere—the migrating diurnal tide, also known as the diurnal period sun-synchronous tide—directly responds to thermal forcing of the zonally averaged component of dust, which is closely dependent on dust storm activity[3,18]. The migrating diurnal tide can cause strong diurnal-varying thermal and dynamic behaviors[3,26–28]. Strong evidence has shown that the vertical wave pattern of water ice in the atmosphere is modulated by the migrating diurnal tide[12,26,28]. However, the global dust diurnal variation and its relationship with the atmospheric migrating diurnal tide during major dust storms have not been previously observed.

Here, using high-temporal-resolution data from the Mars Climate Sounder[29,30] (MCS) onboard Mars Reconnaissance Orbiter, we report that the strong diurnal variation in dust in the southern hemisphere can be explicitly related to the dynamic process of migrating diurnal tide during major dust storms on Mars. We used the retrieved temperature, dust opacity ($d_z\tau$ in units of $km^{-1}$), pressure, and altitude from the MCS between Mars Year (MY) 28 and MY 34. The new cross-track observational strategy of MCS increases the local-time coverage of the measurements[18] and allows us to isolate the zonally averaged diurnal temperature and dust fields in a reference frame with observations at fixed local solar times. Under dust storm conditions, the vertically trapped component[3,18,24] of the migrating diurnal tide both in temperature and dust fields at mid-to-high latitudes is dominant and readily isolated in the observations. The use of classical tidal theory[24,31] allows the associated horizontal velocity fields to be synthesized. In addition, the Mars Climate Database version 5.3[32] (MCD 5.3, see "Methods") is used to compare and validate our results.

## Results

**The dust diurnal variation during major dust storms.** In all 12 major dust storms from MY 28 to 34, the zonally averaged dust behavior at mid-to-high latitudes of the Southern Hemisphere exhibited remarkable diurnal variations (Fig. 1), both in the dust opacity at a certain pressure level and the maximum height that the dust can reach. The diurnal variation in the dust mainly occurs between 10 and 100 Pa (~20 to ~40 km, Fig. 1). In contrast, the variation in the dust opacity before and after the periods of major dust storms is much weaker within a sol (Supplementary Fig. 1). At mid-to-high latitudes, the dust opacity generally shows a monotonic decrease with altitude, while at low latitudes, inversion layers may manifest at high altitudes, usually when the dust opacity is less than 0.0005[6].

For convenience, we exclude the detached layers in this study and define the dust height (DH) as the altitude of the dust opacity of 0.0005 to quantify the dust diurnal variation. Compared with the values before the dust storms, the nighttime DH increases by no more than 10%, while the dust opacity in the daytime exhibits a dramatic change (Supplementary Fig. 2). The diurnal variability in DH in all 12 major dust storms from MY 28 to 34 is shown in Supplementary Fig. 3. The day–night DH shows a clear seasonal and interannual variability tracking the dust storm intensity (Supplementary Fig. 3). The day–night DH ranges from 10 km with a duration of ~10° solar longitude ($L_s$) during the A regional storm in MY 30 to ~40 km with a duration of ~20° $L_s$ during the global storm in MY34. The strong DH diurnal variability tends to occur in the Southern Hemisphere, where major dust storms are known to enhance diurnal thermal tides[18]. Outside major dust storms, a DH day–night variability of up to 10 km is not uncommon.

We selected the A regional dust storm in MY 33 to analyse the evolution of the dust diurnal variation at mid-to-high latitudes of the Southern Hemisphere (Fig. 2). The duration of this major storm is ~30 sols ($L_s = $ ~214° to ~233°). This storm is characterized by the increases in the daytime DH by more than 20 km (Fig. 2a), the temperature by ~30 K and the dust opacity from 0 to 0.0025 (Fig. 2c). To further quantify the dust diurnal variation driven by migrating diurnal thermal tide[17,27] (DW1), we applied nonlinear least squares analysis (see "Methods") to obtain the evolutions of DW1 amplitudes and phases in both dust and temperature (Fig. 2b, d). The migrating semidiurnal tide and stationary planetary waves were previously found to be common and strong in the Martian atmosphere[22,27,33], but they are much weaker than DW1 by up to one order of magnitude during major dust storms[18]. The amplitudes of DW1 in both temperature and dust opacity (Fig. 2b) also show similar time evolutions as those in the DH (Fig. 2a), and temperature and dust opacity at ~50 Pa (Fig. 2c). The evolution of dust storms always exhibits rapid development[34] due to rapid dust lifting[35] and expansion[5,36] processes. A sharp increase in the DW1 amplitudes is also shown at the beginning of the storm in response to the rapidly lifted and expanded dust; however, differences in the phase evolutions of the temperature and dust opacity are more dramatic (Fig. 2d). The phase of the temperature is fixed to ~18:00, corresponding to previous simulation results[37]. However, the phase of the dust tide apparently has a dramatic shift from the morning to the evening (~18:00) when the DW1 amplitude of the temperature increases (exceeding 10 K in this case). Overall, the diurnal dynamic process of the dust is strongly altered by the enhanced migrating diurnal tide during dust storms.

**The rapid meridional motion of the dust front.** Taking advantage of the more local-time coverage of observations, we confirm that the dramatic diurnal variation in the dust occurs

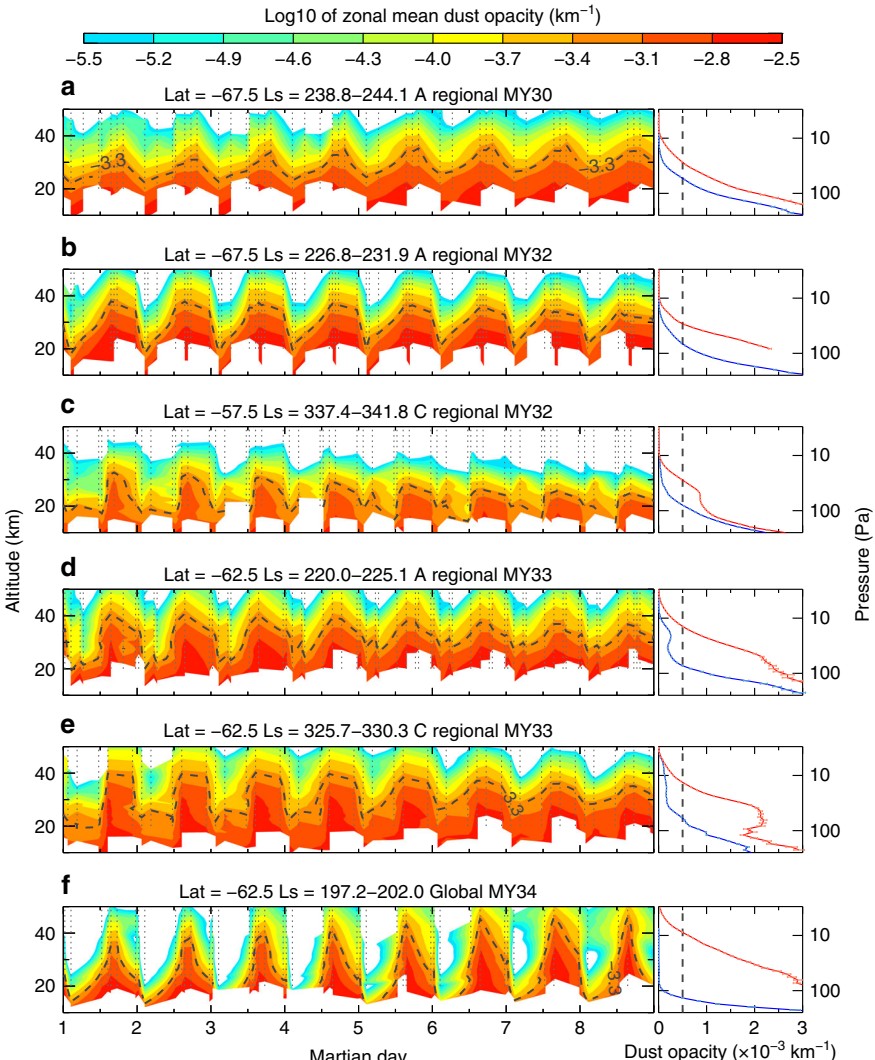

**Fig. 1 Evolutions of the zonal-mean dust opacity at 463 cm$^{-1}$ (22 μm) during major dust storms. a–e** For regional dust storms, and (**f**) for the global dust storm. All cases are obtained during the MCS cross-track observational periods to increase the local-time coverage (Supplementary Fig. 4). The Martian year (MY), $L_s$ and latitude values are labeled at the top of each plot. The 0:00 LTST of each sol corresponds to the marking place of the $x$ coordinate value. Colors are the logarithm of the zonal-mean dust opacity. The retrieval uncertainties are mostly below 6%. At each latitude and pressure level, we zonally average the altitude data ($y$-axis) from MCS. The right panels show the vertical profiles of the dust opacity at 3 a.m. (blue) and 3 p.m. (red) during the corresponding dust storm. The retrieval uncertainties are given by the horizontal error bars (most of them are smaller than the line widths). The black dash contour lines on the left panels and the vertical black dash lines on the right panels represent the dust height (DH) where the dust opacity is 0.0005.

mostly at mid-to-high latitudes because of the rapid meridional transport (Fig. 3). The dust abundance and height at low-to-mid latitudes are enhanced by dust lifting and deep convection[9], while those at high latitudes are not. Consequently, the dust opacity at the mid-to-high latitudes shows a sharp meridional gradient and forms a dust front. The meridional movement of the dust front between ~3 a.m. and ~3 p.m. can cover up to ~15° of latitude during the A regional dust storm (Fig. 3a–c) and ~40° of latitude during global dust storms (Fig. 3d–i, also see refs. [15,16]). The sharpness of the dust opacity gradient depends on the seasonally varying background at high latitudes[34]. For instance, when a global dust storm starts at the southern-summer solstice in MY 28, the background airborne dust has already been widely spread across the Southern Hemisphere (Supplementary Fig. 3). Therefore, the latitudinal dust gradient weakens during the global dust storm in MY 28 (Fig. 3d–f). Nevertheless, the meridional movement of the dust front (defined as where the dust opacity is 0.0005) can cause the same diurnal DH variation revealed in

Fig. 1 and Supplementary Fig. 2. A zoomed-in plot of the high-latitude data with multiple local times illustrates the diurnal meridional motion of the dust front during the A regional dust storm in MY 33 (Supplementary Fig. 6). The dust front moves poleward in the daytime (Supplementary Fig. 6b–e and Fig. 4c, d), reaching the highest latitude (~77° S) at 17:00–20:00, and equatorward in the nighttime (Supplementary Fig. 6a, b, e and Fig. 4c, d), reaching the lowest latitude (~58° S) at 4:00–7:00. The similar dust meridional motion within a sol is also recognized during the global dust storm in MY34[16]. The diurnally meridional movement of the dust mountain extends to the entire Southern Hemisphere (Fig. 3a, b, the movement can cover ~15° of latitude), while its northern counterpart is weaker. The variation in the day–night dust opacity (Fig. 3c) also shows a clear daytime decrease at the low-to-mid latitude band centered at 30° S and a sharp daytime increase at mid-to-high latitudes between 20 and 70 Pa due to the meridional dust transport. The tilted variation structure (Fig. 3c) at the equator and in the Northern Hemisphere

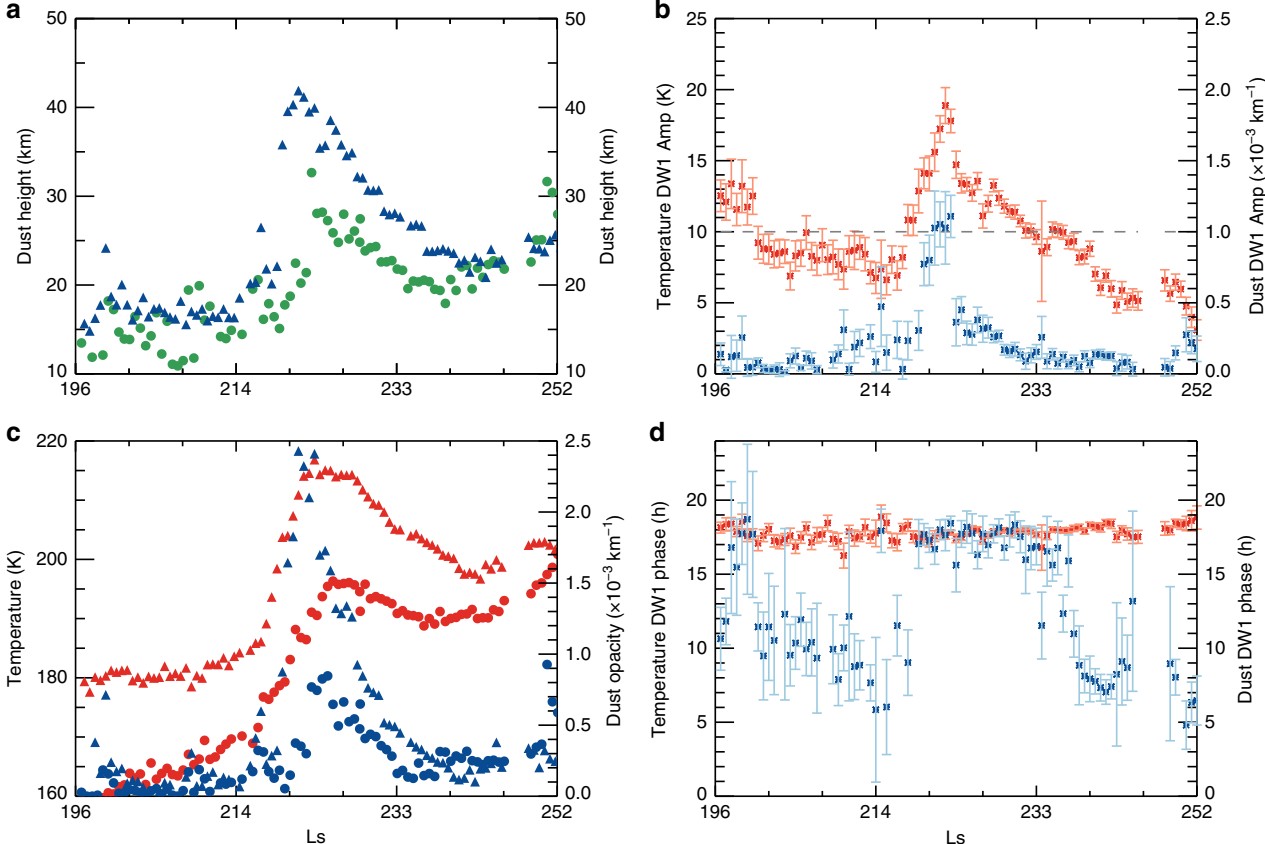

**Fig. 2 Evolution of the dust diurnal variation at 62.5° S during A-season regional dust storm in Mars year 33. a** Dust height corresponding to the zonal-mean dust opacity of 0.0005 (triangles for the daytime and circles for the nighttime). An example of the dust height diurnal variation with multiple local times during the peak time of dust storm is illustrated in Supplementary Fig. 5. **c** Temperature (red) and dust opacity (blue) at ~50 Pa (triangles for the daytime and circles for the nighttime). Amplitudes (**b**) and phases (**d**) of the temperature DW1 (red) and dust opacity DW1 (blue) at ~50 Pa, which is within the main altitude range of the dust diurnal variation. The period runs from $L_s = $ ~196° to ~252° (90 sols, 10 sols for each tick interval of x-axis). Uncertainties in (**a**, **c**) are less than 5%. The vertical error bars in (**b**) and (**d**) indicate the 1-sigma confidence level.

indicates that complicated vertical motion processes such as deep convection, slope wind, tidal vertical wind, and sedimentation become more significant than those at southern mid-to-high latitudes.

**Dust distribution driven by thermal tides.** As an inert and globally distributed species in the Martian atmosphere, dust is one of the best tracers for wind tracking. Rapid meridional motion of the dust front implies strong diurnal meridional winds during major dust storms. Regarding tides as low-frequency gravity waves, we estimated the 3-D wind tides, including their amplitudes and phases, from the temperature tides using a polarization relationship[31]. To investigate the meridional dust transport by horizontal winds, we combined the derived DW1 zonal and meridional winds and the zonal gradient wind (see "Methods") averaged between 20 and 70 Pa during the peak period of the dust storm and produced a 2-D horizontal wind field in one sol (Fig. 4a, b; also see the horizontal wind vectors in Fig. 4c, d). The phase of the DW1 meridional wind is around midnight, indicating a poleward flow during the daytime and equatorward flow during the nighttime. The same pattern of diurnal-varying horizontal wind distribution during the A regional dust storm is also evident in the MCD[32] simulation results (Supplementary Fig. 9). In general, we see a good agreement in both speed and direction between the classical-theory-derived and MCD-simulated winds in the latitude range of 30–70° S, except that our derived wind speeds are

lower than the simulated results around noon and midnight. We also expect that the derived winds have larger biases against the simulated winds at tropics, typically equatorward of 30° S. This difference is because the vertically propagating characteristics in the tropics cause the DW1 to be strongly influenced by the zonal-mean zonal winds, which are not well represented in idealized tidal theory[24,28,38].

We then performed a Lagrangian analysis of particles by tracing the motions of initialized dust particles in the time-varying wind field (see "Methods"). The result shows that the wave-like dust front and its diurnal variation occur at the highest latitudes at ~18:00 and at the lowest latitudes at ~06:00 (Fig. 4c, d). The meridional movement can cover 10–20° of latitude. In addition, our simulations are able to reproduce the negative day–night dust difference in the mid-latitude band centered at 30° (Fig. 3c). In summary, the diurnal variation in the dust front and dust mountain can be well explained by our derived DW1 tidal winds. Our discovery provides the first direct observational evidence that the dynamic processes of the enhanced thermal tides during dust storms strongly influence the dust distribution itself. In return, the associated heat transport by the moving dust from the middle to the high latitudes in the daytime can help increase the day–night temperature contrast of the atmosphere and further enhance the diurnal thermal tide. This finding suggests a strong positive feedback process in the tidal-driven dust movement.

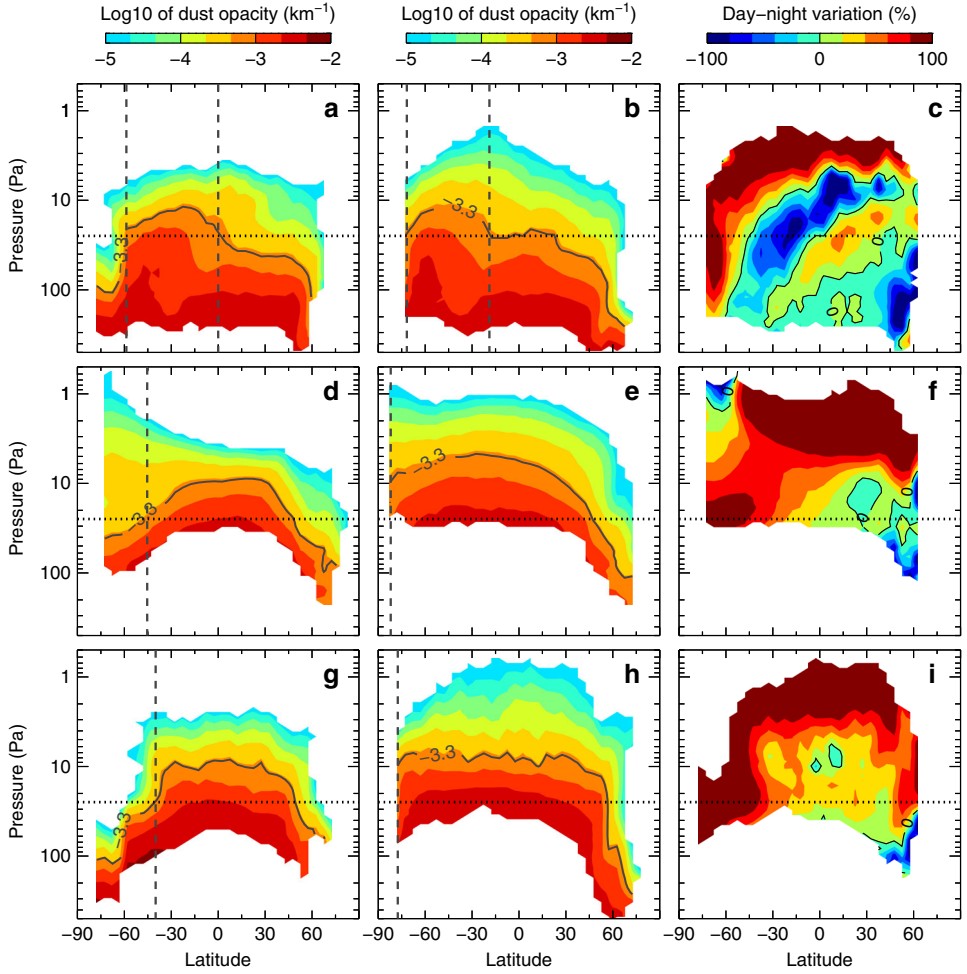

**Fig. 3 Meridional and vertical structures of the zonal-mean dust opacity during the peak time of major dust storms. a–c** $L_s = 220-221.9°$ for A-season regional dust storm in MY 33; **d–f** $L_s = 270.9-272.9°$ for the global dust storm in MY 28; **g–i** $L_s = 206.5-208.4°$ for the global dust storm in MY 34. **a**, **d**, **g** Nighttime at ~3 a.m. (2.5–3.5 a.m. averaged). **b**, **e**, **h** Daytime at ~3 p.m. (2.5–3.5 p.m. averaged). **c**, **f**, **i**: The relative day–night variations in percentage normalized by the daytime opacity. The $L_s$ periods in **a–c** correspond to the peak time of the dust storm in Fig. 2. For (**a**, **b**, **d**, **e**, **g**, **h**), the black solid contour lines indicate the dust height with the opacity of 0.0005. The horizontal dot lines at 25 Pa are used to determine the latitude of the dust front, which is marked by the vertical dash lines. The dust mountain depicts the mountain-like dust distribution between the two dash lines in (**a**, **b**).

## Discussion

The good agreement between the derived wind fields based on the classical theories and the numerical simulations from MCD has two implications. First, the classical gradient and tidal theories are valid for qualitatively analysing the dust tides in the mid-latitudes on Mars and shed light on the underlying physical mechanisms. Second, the dominant dynamic process for the meridional motions of airborne dust between ~10 and 100 Pa in the Southern Hemisphere during major dust storms is westward-propagating migrating diurnal tide. Furthermore, extreme weather events (such as major dust storms) are useful for isolating typical physical processes (such as tidal processes) for analysis. Note that the focus of this study is the behavior of the dust at high altitudes (pressure less than 100 Pa). Dust motions near the surface could be in the opposite direction since lower-level and upper-level tidal winds are out of phase[39,40].

In addition to the airborne dust, the simulated water vapor results from MCD[32] also manifest intense meridional motion within a sol driven by the DW1 tidal wind during major dust storms (Supplementary Fig. 10), which further suggests a broader impact of the rapid meridional motion on the gas species. The global distribution of water vapor shows clear seasonal variabilities[41], implying that the effect of meridional motion on water

vapor might be different in different seasons. The water vapor mixing ratio tends to maximize at the middle and lower latitudes during the A regional dust storms (Supplementary Fig. 10a, b) in the southern-spring-season, when the dayside tidal wind can transport water vapor from the lower to the higher latitudes. In contrast, during the southern-summer-season, the water vapor concentration in the southern high latitudes is dramatically increased due to sublimation of polar cap water ice[42,43]. As such, the nightside tidal wind in this season transports water vapor from the higher to the lower latitudes, but in a limited range under no-dust-storm conditions (Supplementary Fig. 10c). However, during global dust storms, the rapid meridional motion can cover up to 40° latitudes (Fig. 3), corresponding to a mean meridional wind amplitude of ~86 m s$^{-1}$. This suggests a very rapid exchange of heat and materials between the polar region and the middle latitudes. This effect may also have caused the water vapor transport during the global dust storm in the southern summer of MY 28 (Supplementary Fig. 10d), when the nightside tidal wind transported water vapor from the high latitudes all the way to the low latitudes.

Meridional advection of dust and water vapor by tidal wind is expected on a diurnal time scale. The net transport of these fields must be accomplished by circulation elements other than the

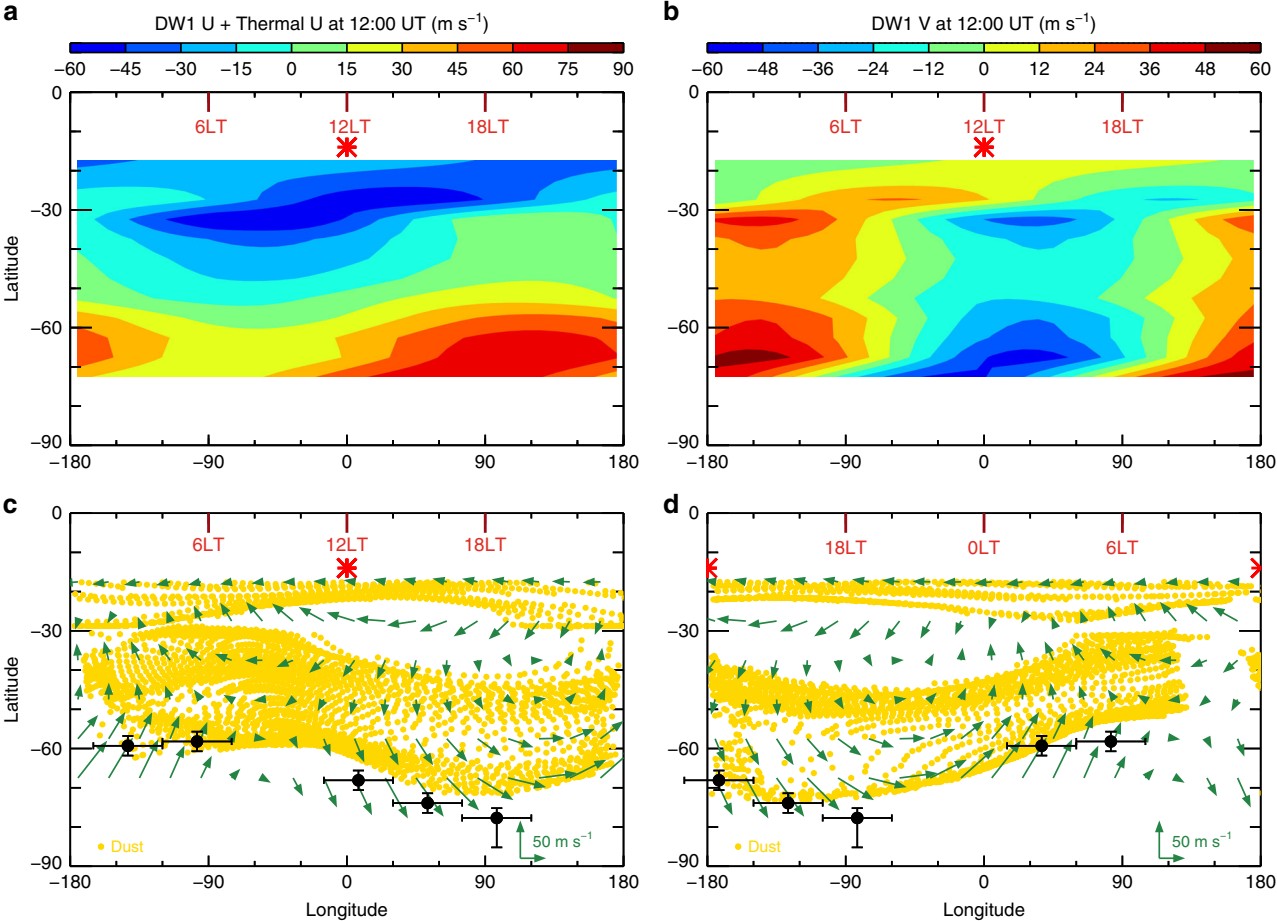

**Fig. 4 Diurnal variation of the dust distribution in the horizontal plane simulated using the Lagrangian particle analysis.** The reconstructed zonal (**a**) and meridional (**b**) winds are derived from zonal gradient wind plus DW1 winds and averaged between 20 and 70 Pa at 12:00 universal time (UT), $L_s$ = 220–221.9°. The phases of U and V of the DW1 winds are consistent with those shown in Supplementary Fig. 8. The virtual dust particle distributions (yellow points) over-plotted with the horizontal wind vectors (arrows) at 12:00 UT and 24:00 (00:00) UT, shown in (**c**, **d**), respectively, are extracted from the Lagrangian particle analysis. The black dots with error bars in (**c**, **d**) show the latitude values of the observational dust front (see Supplementary Fig. 6) at corresponding local times. The red ticks indicate the local times (LT). From **c**, **d**, the sun (red star symbols) propagates westward from 0° to 180° longitude, along with the wind field pattern and the wave-like dust distribution.

regularly oscillating thermal tide, e.g., the slowly retrograde propagating Rossby waves[44] and the deep convective mesoscale circulations in the southern mid-latitudes in the early stages of major dust storms[45]. These elements can help break the periodic tidal wind loop and induce the net meridional transport of atmospheric materials. Simulations suggested that thermal tides can enhance the net meridional transport of aerosols through momentum flux divergence via tidal dissipation[46]. According to recent observational and numerical studies, the dusty deep convection becomes more widespread in the mid-latitudes under dust storm conditions[45,47]. It should be noted that in the period from noon to the early afternoon, when dusty deep convection is expected to be the strongest, the water vapor is retreating to higher latitudes from the equatorial region (Supplementary Fig. 10d). However, a considerable amount of water still remains in the mid-to-low latitudes (20–60° S latitude and −180° to −150° longitude in Supplementary Fig. 10d), which makes it highly susceptible to be lifted by deep convection. As such, the deep convection occurred in the mid-to-low latitudes could effectively break down the tidal wind loop and induce the net transport of water vapor from the polar reservoir to the lower latitudes during southern-summer-season dust storms. More observations of the diurnal variation in global water vapor and sophisticated model studies incorporating the latest discovery of

rapid dynamic processes such as deep convection are required to better understand this rapid dynamic process in the Martian atmosphere.

Extratropical Rossby wave activity on Mars is weaker in the summer than in the winter and stronger in the Northern Hemisphere than in the Southern Hemisphere[33,48,49]. Therefore, one would not expect strong mid-to-high-latitude exchange in the southern hemisphere during spring and summer. However, the Martian thinner atmosphere increases the power of thermal tides, especially extratropical tides, during dust storm activity[18,50]. Our work suggests a potential mechanism that allows atmospheric water content to be rapidly transported from the polar source[42,51] to the low and middle latitudes by the nighttime tidal wind and then lifted by daytime deep convection[9] and further transported to the Northern Hemisphere by the intensified meridional circulation[52]. This may contribute to the rapid enhancement of water content at high altitudes and hydrogen escape during the southern-summer-season global dust storms, such as that observed in MY 28[9].

## Methods
**MCS dataset**. The MCS instrument[30] on board the Mars Reconnaissance Orbiter has measured the thermal emissions of the Martian atmosphere via limb and on-planet views since September 2006 with a vertical resolution of ~5 km from near

the surface to ~80 km between ~85° and ~85° N. As in a polar sun-synchronous orbit, the MCS scans the Mars atmosphere at ~3 a.m./3 p.m. local true solar time (LTST) with a latitudinal spacing of 110 km (1.86°) and an orbit-to-orbit longitudinal spacing of 27° in 13 orbits within each Martian day (sol). The new cross-track observational strategy can observe ~2 h before and after the nominal 3 a.m./3 p.m. LTST in the low latitudes[22], along with the along-track and off-track observations, making it possible to acquire six local times in the low and middle latitudes and seven or more local times in the high latitudes. Note that the six local-time observations per sol in the lower latitudes are very unevenly spaced, and the error estimates of the semidiurnal harmonic and higher order harmonics are quite large[22]. However, the new observational strategy provides an outstanding observational constraint on the diurnal temperature and aerosol cycles, especially in the mid-to-high latitudes. With the cross-track data of MCS, we can directly investigate the dust opacity at different local times within a sol or perform nonlinear least squares fitting and evaluate the amplitude and phase of the diurnal variation.

We used version 5 of the MCS dataset[53] available from the Planetary Data System at https://atmos.nmsu.edu/data_and_services/atmospheres_data/Mars/Mars.html. MCS retrievals are provided at 105 vertical pressure levels. The corresponding altitude data (with an uncertainty of 1 km), which is obtained based on the geometric pointing of the instrument, are also gridded via pressure coordinates[14]. These altitude data from MCS are used to determine the diurnal variation in DH. The MCS dataset used in this work is from $L_s = 111°$ in MY 28 to $L_s = 269°$ in MY 34, including all in-track and cross-track observational data. The cross-track observational data in four MYs (MYs 30–34) are obtained intermittently (Supplementary Fig. 4), except for only one completed year in MY 33. The entire durations of the A and C regional dust storms in MY 33 and the global dust storm in MY34 are observed. However, the MCS retrievals have limited vertical and local-time coverage in the case of high opacity[54], reducing the data quality during the global dust storm in MY 34. As such, we selected the A-season regional dust storm in MY 33 to investigate the evolution of the diurnal variation in the dust at the southern mid-to-high latitudes.

**Major dust storms.** Conventionally, major dust storms refer to storms that dramatically influence global-scale atmospheric thermal and dynamic structures and circulations[4,34], including planet-encircling global storms and three types (A, B, and C) of large regional dust storms during the dusty seasons[34]. All three types of regional dust storms are defined in ref. [34]. A-season regional storms are planet-encircling southern hemisphere events occurred at springtime. B-season regional storms are southern polar events and usually occur near the perihelion and last through the solstice. C-season regional storms are southern summertime events and start well after the end of B-season regional storms. A and C storms occur in the Southern Hemisphere but have northern responses due to the intensification of the overturning Hadley circulation driven by the enhanced thermal tides[46,55] and direct dust heating[34]. However, B storms have limited dynamic effects and are not considered in this work. A and C storms are typically the same type of storm, but C storms are weaker and occur more irregularly than A storms. In this paper, when major dust storms are mentioned, only A and C regional storms and global dust storms are considered. The detailed occurrence time and region for the A and C regional dust storms are listed in ref. [34]. The A and C regional dust storms and global dust storms usually originate as flushing storms that cross the equator into the Southern Hemisphere. The dust activity and dynamic responses at the onset of dust storms can be quite localized[5,36]. However, the dust column opacity and mid-level (50 Pa) temperature responses of these events span a broad range of latitudes after the onset[34].

**Zonal averages.** The data (temperature, dust opacity, and altitude) were first divided into different periods under different conditions, e.g., one $L_s$ bin interval for Supplementary Fig. 2, a running window of three $L_s$ bin intervals for Supplementary Fig. 3 to show long-term variations, or one sol bin interval for Fig. 2 to describe the detailed evolution of a dust storm. Then, for each period, the data were further binned into local-time-dependent 3-D arrays corresponding to the 3-D space in terms of latitude, longitude, and pressure level. Considering the temporal and spatial data coverage and resolution of the MCS described in the dataset section above, we used 1-h local time, 10° latitude and 30° longitude bin intervals for each pressure level. Finally, we obtained a $24 \times 18 \times 12 \times 105$ (local time × latitude × longitude × pressure level) array. The longitude coverages of ~3 a.m. and ~3 p.m. are the most complete due to the dominant in-track observation strategy[22]. The zonal averages of these local times were calculated by averaging the longitude bins at each certain latitude and pressure level with no less than 10 of the total 12 longitude bins to ensure an even longitudinal weight. To meet this criterion, we applied a running window of four sols to increase the longitudinal coverage for Fig. 1 and Supplementary Figs. 1 and 5. The other local times observed by the cross-track or off-track strategies have less longitude coverage due to the relatively fewer observational times. The zonal averages of these local times were created by rebinning multiple local times (3 h) together to increase the longitude coverage and then averaging the longitude bins with no less than 8 of the total 12 longitude bins (e.g., Supplementary Fig. 6). A more detailed description of this data-binning strategy and the uncertainty analysis is shown in Supplementary Fig. 7.

**Tidal component fitting.** Atmospheric tides can be regarded as zonally and vertically propagating waves with the form[17,26]

$$\varphi = \sum_{\sigma,s} \varphi^{\sigma,s}(\theta,p)e^{i(s\lambda - \sigma t)}, \tag{1}$$

where $t$ and $\lambda$ are the universal time (the local time at 0° longitude) and longitude; $\sigma$ and $s$ indicate the frequency and zonal wave number for a certain wave (e.g., $\sigma = 1$ and $s = -1$ for the westward-propagating diurnal tide with zonal wave number 1, referred to as DW1, where D means diurnal, W means westward propagating and 1 means zonal wave number 1); $\theta$ and $p$ are the latitude and pressure level; $i$ is an imaginary unit; and $\varphi$ can be the temperature, wind, geopotential, etc. of the atmosphere perturbations. Here, we performed a 2-d (in local time and longitude) nonlinear least squares fitting of the following transformed expression of Eq. (1)[38]:

$$T(\lambda,\theta,p,\hat{t}) = \sum_{\sigma,s} \left(C^{\sigma,s}(\theta,p)\cos\left((s+\sigma)\lambda - \sigma\hat{t}\right) + S^{\sigma,s}(\theta,p)\sin\left((s+\sigma)\lambda - \sigma\hat{t}\right)\right), \tag{2}$$

where the local time in the satellite coordinate $\hat{t} = t + \lambda$ was substituted for $t$. The wave modes we chose for the fitting are the diurnal migrating tide ($\sigma = 1$, $s = -1$), the semidiurnal migrating tide ($\sigma = 2$, $s = -2$), and the stationary planetary wave with zonal wave number 1 ($\sigma = 0$, $s = 1$). We used the binned 4-d array described in the above section but with a smaller longitude bin interval (10° longitude) to increase the longitude resolution to increase the fitting accuracy. Then, the amplitude and phase of DW1 at a certain latitude $\theta$ and pressure level $p$ can be obtained by $\sqrt{C^{1,-1}(\theta,p)^2 + S^{1,-1}(\theta,p)^2}$ and $\tan^{-1}(C^{1,-1}(\theta,p)/S^{1,-1}(\theta,p))$.

The fitting uncertainty estimates of both the amplitude and phase are presented by 1-sigma confidence intervals to represent the goodness of fit. Note that while the diurnal migrating tide can be obtained with good confidence based on this procedure, the semidiurnal tide and the stationary wave can be significantly aliased at low latitudes because of the uneven local-time coverage described above in the MCS dataset section. The above procedures are performed for MCS temperature retrievals for the temperature DW1 tide, and dust opacity retrievals for the dust DW1 tide.

**Zonal gradient wind.** The estimated daily and zonally averaged zonal wind (gradient wind) $\hat{U}(p)$ at a certain latitude $\phi$ and pressure level $p$ is calculated by integrating the thermal wind equation[12,56] from the lowest pressure level of temperature retrievals $p_0$ with the assumption of no motion in the pressure level $p$:

$$\hat{U}(p) = \int_{p_0}^{p} \frac{R}{f} \left(\frac{dT}{dy}\right)_{p'} d\ln p', \tag{3}$$

where $R$ is the specific gas constant, $f$ is the Coriolis parameter for the latitude $\phi$, and $\left(\frac{dT}{dy}\right)_p$ is the daily and zonally averaged temperature gradient in the meridional direction at a certain pressure level $p'$. The daily and zonal averaged temperature is computed by averaging the zonal-mean temperature of local time 3 a.m. and 3 p.m. Interpolation is applied properly when calculating the temperature gradient.

**Tidal wind.** No wind measurements of the Martian atmosphere, especially of the time-varying global-scale wind fields, are available. However, polarization relationships of tides in a dissipation-less atmosphere have been derived by regarding tides as low-frequency gravity waves[31,57], which provides the possibility to compute the wind tides from the temperature tides in an ideal condition with several assumptions to investigate the daily variation in the wind fields.

Here, we briefly summarize this method. Based on the assumptions of dissipation-less, zero-mean wind, and no horizontal temperature gradient, the primitive equations[25] for large scale wave perturbations such as tides can be expressed as a set of linearly coupled equations in log-pressure coordinates ($z = -H_s \ln(p/p_s)$, where $p$, $p_s$, and $H_s$ are pressure, pressure at a reference altitude, and scale height at a reference altitude, respectively):

$$u'_t - fv' + (a\cos\phi)^{-1}\Phi'_\lambda = X', \tag{4a}$$

$$v'_t + fu' + a^{-1}\Phi'_\phi = Y', \tag{4b}$$

$$(a\cos\phi)^{-1}[u'_\lambda + (v'\cos\phi)_\phi] + \rho_0^{-1}(\rho_0 w')_z = 0, \tag{4c}$$

$$\Phi'_{zt} + N^2 w' = \frac{\kappa J'}{H_s}, \text{ where } N^2 = \frac{R}{H_s}\left[\frac{\partial T_0}{\partial z} + \frac{\kappa T_0}{H_s}\right], \tag{4d}$$

$$\Phi'_z = \frac{\partial \Phi'}{\partial z} = \frac{RT'}{H_s}, \tag{4e}$$

where $u'$, $v'$, $w'$, $T'$, and $\Phi'$ are the perturbations in the zonal wind $u$, meridional wind $v$, vertical wind $w$, temperature $T$, and geopotential $\Phi$; the subscripts except 0 and s indicate partial derivatives; the subscript 0 in $\rho_0$ and $T_0$ denotes the background state $A_0 = A - A'$ ($A = \rho, T$) and indicates the daily and zonal averages in this work; $t$, $\phi$, $\lambda$, $\Omega$, $a$, $\rho$, and $R$ are the time, latitude, longitude, rotation rate,

mean radius, density, and specific gas constant on Mars; $\kappa = R/c_p$ where $c_p$ is the specific heat at a constant pressure; $N$ is the buoyancy frequency; and $f$ is the Coriolis parameter, where $f = 2\Omega \sin \phi$. The terms $X'$, $Y'$, and $J'$ represent the dissipations of the zonal and meridional wind perturbations and the diabatic heating forcing, respectively.

We are concerned about the polarization relations among $u'$, $v'$, $w'$, $T'$, and $\Phi'$, which can be expressed for a certain wave mode $(s, \sigma)$ as:

$$(u',v',w',T',\Phi') = \mathrm{Re}\left\{ \left[\tilde{u}(\phi,z), \tilde{v}(\phi,z), \tilde{w}(\phi,z), \tilde{T}(\phi,z), \tilde{\Phi}(\phi,z)\right] e^{i(s\lambda - \sigma t)}\right\}, \quad (5)$$

where Re means the real part of the expression.

$\tilde{u}(\phi,z)$, $\tilde{v}(\phi,z)$, $\tilde{w}(\phi,z)$, $\tilde{T}(\phi,z)$, and $\tilde{\Phi}(\phi,z)$ are complex quantities in terms of $\phi$ and $z$ and include the information of their amplitudes and phases. To obtain the ratios of these five amplitudes, only four homogeneous linear equations are needed. As ref. [31] suggested, we found it convenient to use Eqs. (4a–4e) to relate the complex fields $\tilde{u}$, $\tilde{v}$, $\tilde{w}$, and $\tilde{T}$ in terms of the complex geopotential $\tilde{\Phi}$ by substituting Eq. (5) into these equations and assuming $X' = Y' = J' = 0$. Then, we have the following:

$$-i\sigma\tilde{u} - f\tilde{v} + (a\cos\phi)^{-1} \; is \; \tilde{\Phi} = 0, \quad (6a)$$

$$-i\sigma\tilde{v} + f\tilde{u} + a^{-1}\tilde{\Phi}_\phi = 0, \quad (6b)$$

$$\tilde{w} = \frac{i\sigma}{N^2}\tilde{\Phi}_z, \quad (6c)$$

$$\tilde{T} = \frac{H_s}{R}\tilde{\Phi}_z. \quad (6d)$$

From Eqs. (6a, 6b) we can relate $\tilde{v}$ and $\tilde{u}$ by

$$\tilde{v} = \frac{-i\sigma\tilde{\Phi}_\phi - f(\cos\phi)^{-1} \, is \, \tilde{\Phi}}{f\tilde{\Phi}_\phi - i\sigma(\cos\phi)^{-1} \, is \, \tilde{\Phi}}\tilde{u}. \quad (7)$$

From Eqs. (6c, 6d) we can relate $\tilde{w}$ and $\tilde{T}$ by

$$\tilde{w} = i\frac{R\sigma}{H_s N^2}\tilde{T}. \quad (8)$$

For a westward-propagating monochromatic wave, $\tilde{\Phi}_\phi = 0$. Then, Eq. (7) is reduced to:

$$\tilde{v} = -i\frac{f}{\sigma}\tilde{u}, \quad (9)$$

which is the polarization relation of low-frequency gravity waves[57]. Note that $\tilde{\Phi}$ is related to $\tilde{T}$ in terms of the vertical wave number $m^{s,\sigma}$ for a certain wave mode $(s, \sigma)$ by[31]

$$\tilde{\Phi} = \frac{R}{H_s}\left[im^{s,\sigma} + \frac{1}{2H}\right]^{-1}\tilde{T}. \quad (10)$$

We can then relate $\tilde{T}$ to $\tilde{u}$ by substituting Eqs. (9 and 10) into Eq. (6a):

$$\tilde{T} = \frac{H_s}{R}\left[im^{s,\sigma} + \frac{1}{2H}\right]\frac{\sigma^2 - f^2}{\sigma k}\tilde{u}; k = \frac{2\pi s}{2\pi a\cos\phi}. \quad (11)$$

Finally, we have Eqs. (9 and 11), from which the wind fields $\tilde{u}$ and $\tilde{v}$ can be computed from the known structure of $\tilde{T}$.

Using the MCS temperature dataset and the nonlinear least squares fitting process, we obtained the $\tilde{T}$ structures of temperature DW1 in both amplitude and phase. The vertical wave number $m^1$ can also be estimated from the vertical structure of the DW1 phase.

Here, we show the computed amplitudes and phases of $\tilde{T}$, $\tilde{u}$, and $\tilde{v}$ averaged from 20 to 70 Pa during the A regional dust storm of MY33 in Supplementary Fig. 8. A distinct increase in the temperature, zonal wind, and meridional wind amplitudes can be found during the peak of the dust storm in the southern mid-to-high latitudes. The phase (the local time of the maximum northward wind in the southern mid-to-high latitudes) of the meridional wind is near midnight during the dust storm. Note that the vertical wavelength at mid-to-high latitudes is large (relatively uniform in phase change with height) between 100 and 10 Pa, indicating a vertically trapped wave pattern[17,24], while in the tropics (typically equatorward of 30°), the vertical wavelength is finite (25–30 km when the dust amount is low), and the phase change becomes complicated during dust storms; this phase change may be strongly influenced by the enhanced zonal-mean winds[28]. Therefore, the wind results in the tropics may not be properly represented in this idealized tidal wind model.

**Lagrangian particle analysis.** First, we constructed a 24-h long, 2-D horizontal wind field during the peak sols (averaged to one sol) of the considered dust storm (corresponding to the period shown in Fig. 3a–c) by combining the estimated zonal gradient winds and the DW1 tidal winds. The meridional and vertical structure of the estimated zonal gradient winds is similar to that suggested by previous work in the same season[12] (data not shown), with eastward wind in the southern mid-to-

high latitudes and westward wind in the low latitudes. The gradient wind calculated near the tropics where the Coriolis parameter approaches zero is neither plotted nor used. The zonal gradient wind is a daily average and regarded as the background. Then, the time-varying 2-D DW1 tidal winds are reconstructed by substituting the amplitude and phase of the DW1 zonal and meridional winds of each latitude and the pressure level back into Eq. (2). The DW1 zonal wind is then added to the zonal gradient wind background to form the total zonal wind field (Fig. 4a). The meridional wind field is approximate to the reconstructed DW1 meridional winds (Fig. 4b). The total horizontal wind field is obtained by performing the vector addition of the total zonal winds and meridional winds (vectors in Fig. 4c, d). All the winds described above are vertically averaged between 20 and 70 Pa corresponding to the main dust front vertical range. Note that from the discussion on the potential biases of wind results in the tropics in the above section and Supplementary Fig. 9, the derived DW1 meridional wind field in the mid-to-high latitudes is validated by the MCD simulation, while in the tropics (equator-ward of 30°), it differs significantly from that of the MCD. Therefore, the results based on the derived wind field in the tropics should be treated with caution.

In the analysis, we increase the horizontal resolution of the wind field with a grid interval of 2.5° latitude × 5° longitude by interpolation. At the beginning of the procedure, a large set of virtual dust particles are initialized uniformly in the range of −180° to 180° longitude and 17.5–72.5° S latitude with a grid interval of 1° latitude × 5° longitude. Then, the particles are integrated within the 2-D, time-evolving wind velocity fields with a timestep of 1 s. The total integrating time of the procedure is three sols, and each sol has the same wind field duplicate. The spin-up time is within ~6 h, after which the dust distribution is mostly repeatable at the same local time of different sols.

**MCD version 5.3.** MCD version 5.3[32] (MCD 5.3) is an advanced database of meteorological fields built with output from the Laboratoire de Météorologie Dynamique General Circulation Model[58] (LMD GCM) and validated using the latest observational data from the Thermal Emission Spectrometer, Thermal Emission Imaging System, MCS, and Mars Exploration Rovers, etc. MCD 5.3 can provide all the primary meteorological fields, such as temperature, wind, density and atmospheric composition, including dust, water vapor, and water ice content, as the LMD GCM includes the full dust cycle[59] and water cycle[60]. Version 5.3 provides 8 Martian Year scenarios from MY 24 to MY 32 using specific dust loading and solar EUV input for these years. The horizontal wind and water vapor data during periods of the global dust storm of MY 28 and the A regional dust storm of MY 32 from MCD 5.3 are used in this work. MCD 5.3 provides no results for MY 33, so we use MY 32 instead for comparisons between the derived and simulated wind fields during the A regional dust storm shown in Supplementary Fig. 9. Note that the A regional dust storms in MY 32 and MY 33 have similar occurrence times (Supplementary Fig. 3).

## Data availability
The MCS data used in this study are freely available for download at https://atmos.nmsu.edu/data_and_services/atmospheres_data/Mars/Mars.html. The MCD data are available at http://www-mars.lmd.jussieu.fr.

## Code availability
Code used in this study can be obtained from the corresponding authors upon request.

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

## Acknowledgements

This work is supported by the B-type Strategic Priority Program of the Chinese Academy of Sciences, Grant No. XDB41000000 and the pre-research project on Civil Aerospace Technologies of China National Space Administration, Grant No. D020105. Z.W., T.L., and J.C. acknowledge supports from the National Natural Science Foundation of China through grants 41525015, 41774186 to J.C. and 41674149 to T.L., X.Z. acknowledges support from NSF grant AST1740921. Z.W. is also supported by the grant from Key Laboratory of Lunar and Deep Space Exploration, Chinese Academy of Sciences (LDSE201803). We would like to thank the MCS and MCD teams for making both datasets available online.

## Author contributions

Z.W. conceived the study. Z.W., T.L. and X.Z. designed the study and contributed to the scientific discussions. Z.W. designed and analyzed the dust tide diagnoses and Lagrangian particle analysis. T.L. and J.L. contributed to the tidal wind derivation. J.L. contributed to the data analysis of the Mars Climate Database and the preparation of the supplementary figures. T.L., X.Z. and J.C. assisted Z.W. with the preparation of the paper.

## Competing interests

The authors declare no competing interests.
