## [Peer Review File · Nature Communications]

Reviewers' comments:

Reviewer #1 (Remarks to the Author):

In this manuscript, Wu et al. demonstrate that during major dust storms on Mars, a volume of high dust concentrations (or "dust front") migrates westward along with the Sun in line with the main westward-migrating diurnal tide. Mirroring this dust front, less dusty air migrates westward opposite to the Sun on the nightside. The conclusion drawn by the authors is that the dust front originates from rapid meridional transport at 10-40 km altitude (winds of almost 90 m/s) from dusty air in the mid-latitudes. In that case, less dusty and presumably moist air from the polar cap is likely being rapidly transported on the nightside to the mid-latitudes. Such a water supply, if entrained by dusty deep convection at low-mid latitudes, will effectively bring water from the polar cap to high in the middle atmosphere, where it can be photodissociated and enhance the supply of hydrogen that can escape from Mars's atmosphere.

The manuscript is awkwardly written and the figures are not always as impactful as they should be. That said, the manuscript is well-organized, well-argued, and methodologically expansive. Its claims should be of major interest to the Mars atmospheric community and anyone who studies the long-term evolution of Mars's climate. I therefore think the manuscript should be suitable for publication in Nature Communications after revision.

At this point, I should note that the novelty of the claims may not appear so novel by the time it is published. Armin Kleinböhl and Aymeric Spiga in particular have been studying this phenomenon within the 2018 global dust storm: <http://adsabs.harvard.edu/abs/2018AGUFM.P43J3872K> and <https://www.hou.usra.edu/meetings/ninthmars2019/pdf/6146.pdf>

And it is my understanding that a relevant manuscript is under review.

However, this manuscript surpasses any competing manuscript in terms of the scope of the dust storms analyzed and the elegance of the demonstration of the association between the DW1 tidal mode and the dust front.

As it stands, the manuscript makes three major claims:

1. There are rapidly migrating dust fronts during major dust storms on Mars.
2. These dust fronts are driven by the westward migrating diurnal thermal tide.
3. These fronts will transport moist air from the summer pole to lower latitude.

The first claim is supported by Figure 1 (along with Supplementary Figure 1), but Supplementary Figure 1's format is different enough from Figure 1 to undermine the argument for the general reader. The Supplementary Figure needs to show examples of what 10 days of diurnal change in dust opacity would look like prior to commencement of the dust storms in Figure 1.

The second claim is elegantly supported by Figure 2.

The third claim is partly supported by Figure 4 and Supplementary Figure 5 but requires a great deal of unreferenced speculative language at lines 168-171 to make it across the line. The manuscript

does not make it clear how exactly water vapor would escape from the tidal trap. Discussion of the third claim also needs to emphasize that the mechanism only works if transport is from the summer pole (always the southern pole in the cases examined by the manuscript). Otherwise, nightside transport would be from air much drier than that at the mid-latitudes. Typical Martian dewpoints are 170-190 K, while winter pole cap temperatures are 140 K.

The recommendation I would make about the third claim is to emphasize that any sufficiently strong circulation at lower latitudes that is asymmetric with the westward-propagating diurnal tide in the southern mid-latitudes will result in net meridional transport. One such circulation (a very slow-moving wave) is suggested by Clancy et al. (2010, doi: 10.1016/j.icarus.2009.10.011). I am loathe to recommend my own work, but Fig. 14a-c,e-g in Heavens et al. (2019, 10.1175/JAS-D-19-0042.1) shows likely examples of deep convective mesoscale circulations in the southern mid-latitudes in the early stages of major dust storms.

I do not want to spend any time copyediting the manuscript, which is a matter, I think, for the authors and the editors of the journal. But I found it easier to understand the manuscript by rewriting the abstract in this way:

"Mars's atmosphere is strongly affected by the spatial and temporal variability of airborne dust.1-4. Observations have shown interannual, interseasonal, and diurnal variability in the dust distribution 5-9. However, the diurnal variation – global dust change within a sol (Martian day) – is still poorly understood10,11. Although short-term dynamic processes are crucial to rapidly transporting dust and water to higher altitudes12,13, their significance is often underestimated in Martian climate models1 due to the lack of diurnal data. Here we report the discovery of ubiquitous, strong diurnal tides of dust in the southern hemisphere of Mars. Using data with a relatively high local time resolution from the Mars Climate Sounder, we found that zonally circled dust fronts of 10-40 km high slosh back and forth in a wide latitudinal range up to 40° (~2300 km) within one sol during major dust storms. We show that these dust fronts are primarily driven by the westward-propagating diurnal thermal tide. Tidal transport of dust in this way would rapidly transport heat and constituents meridionally, allowing moist air near the summer pole to be rapidly transported to lower latitudes at night, where it then can be lifted by deep convection during the day and contribute to hydrogen escape from Mars during global dust storms 12."

Minor comments:

Abstract: I do not think "zonally circled" works well to describe the dust fronts. They are actually oriented in the meridional direction but travel zonally.

Line 35: "is only" rather than "only possesses"

Lines 52-53: This claim about processes is unsupported.

Lines 68-69: Perhaps rephrase to "where major dust storms are known to enhance diurnal thermal tides"

Line 70: Perhaps rephrase to: "Outside major dust storms, the DH day-night variability of up to 10 km is not uncommon."

Line 70: "Previously expected" by whom?

Lines 87-90: Particularly awkward.

Line 99: "has a dramatic shift"

Line 105: "dramatic" rather than "dramatical"

Line 136: "Rapid meridional motion of the dust front implies..."

Lines 172-173: As noted in the Methods, the water content proxy is limited to 30 km altitude and non-dusty atmosphere, primarily tracer of water getting high in the middle atmosphere; it is a bit of a stretch to use it to diagnose transport over the 10-40 km altitude range mentioned in the abstract.

Supplementary Figure 6: The layout is overly confusing. An improvement would be to put the e-h colorbars and labels at the bottom of the figure.

Nicholas Heavens

Reviewer #2 (Remarks to the Author):

Review of manuscript "Dust Tides and Fast Meridional Motions in the Martian Atmosphere During Major Dust Storms" submitted to Nature Communications

This manuscript presents results of analyses of the spatial and temporal variabilities of dust in the Martian atmosphere during global and large regional dust storms. Analyses are based on profile measurements of temperature and dust by the Mars Climate Sounder (MCS) remote sensing instrument. The manuscript illustrates variations in the zonally averaged dust between daytime and nighttime, with daytime dust extending much farther into the southern high latitudes than nighttime dust during major dust storms. The authors use so called cross-track measurements by MCS to extend the local time coverage of their analysis. The authors fit tidal expressions to the temperature and dust data, and use fundamental equations to calculate zonal gradient winds and tidal winds. With these idealized wind fields a Lagrangian particle analysis is performed, in which virtual particles are initialized and tracked as they are being moved by the zonal and tidal winds. The analysis suggests that the diurnal variation of the dust is created by wind fields that are altered according to the westward migrating diurnal tide. In addition, the manuscript presents calculations of total water content based on water ice opacities observed by MCS. It argues for a meridional motion of the total water content during a Martian day and suggests a fast transport mechanism from a polar source to mid-latitudes at night and lofting to higher altitudes during the day, which may lead to enhanced hydrogen escape.

The manuscript describes features in the Martian atmosphere that are likely to be interesting to scientists specialized in the field of atmospheric dynamics. The variability of aerosols, in particular on sub-diurnal time scales, is an area that is understudied in the Martian atmosphere. The manuscript

provides a credible description of the dust variability observed by the Mars Climate Sounder. However, I do have significant concerns about some of the methodologies applied to interpret the observations, which I will describe in more detail below. Some methodologies are not state-of-the-art while at least one seems to be fatally flawed, leading to conclusions that are unsupported and possibly wrong. Due to these concerns I cannot recommend the manuscript for publication by Nature Communications.

One of my points of criticism concerns the treatment of the MCS cross-track data. MCS cross-track measurements were only obtained in campaigns. While the availability of these measurements is displayed in supplementary figure 3, the detailed characteristics of these data does not seem to be considered adequately. The time difference of cross-track measurements to standard measurements along the orbit track varies significantly with latitude, from +/- 1.5 hours at the equator to over +/- 3 hours at higher latitudes. Several parts of the analysis consider 3-hour local time averages (e.g. figure 3, supplementary figures 5, 7). These somewhat arbitrarily selected time windows are likely to contain mixtures of cross-track and along-track measurements, which obscures the actual local time coverage that goes into the averages, potentially causing biases in the results. Such biases have not been discussed at all in the manuscript. In addition, some figures show data that contains cross-track measurements next to data that does not (e.g. the dust storm in Mars year 28 does not have any cross-track coverage, while the storms in Mars years 33 and 34 do). This is likely to cause biases in the comparisons, especially when data across a large range of latitudes are considered. Also, zonal averages at the onset of large-scale dust storms (such as shown in figure 1) may not be very meaningful as dust activity and dynamical responses can be quite localized at the onset of dust storms. No consideration of this is provided in the manuscript.

Another and more substantial point of criticism concerns the methodology of deriving the wind fields for the simulation of their particle trajectories. Zonal winds are derived from gradient winds and are considered constant over the course of the Martian day, while time-dependent winds are derived from tidal wind equations. While both expressions are valid equations, they have severe limitations when applied to real-world meteorological conditions. Gradient winds are only available over a certain range within the mid-latitudes, and require an assumption of “no motion” (line 277) at a certain pressure level, which is not necessarily true. The tidal winds require “dissipation-less, zero mean wind and no horizontal temperature gradient” (line 292-293). Virtually all these assumptions are not fulfilled. Gradient winds have been assumed as mean winds, which are obviously not zero. Horizontal temperature gradients are significant, certainly along longitude circles but even along latitude circles, e.g. due to stationary waves and non-migrating tides. Finally, the Martian atmosphere has a very short radiative time constant, so it cannot be considered dissipation-less. While the authors acknowledge that diabatic heating is not zero (line 337) they argue that this effect would only influence the vertical wind but would have limited effects on horizontal wind structure. No evidence is provided for this statement. In contrast, the authors show vertical wind in supplementary figure 6 and discuss it in the manuscript (lines 352-362).

The derivation of wind fields from fundamental equations using the aforementioned assumptions cannot be considered state-of-the-art. The way to quantify such processes nowadays is through analyses of simulations from General Circulation Models (GCMs). Modern Mars GCMs are well capable of simulating tidal processes such as the ones dealt with in this manuscript. Typically they will have to be driven with a dust climatology as surface lifting of dust is not simulated well enough

but such climatologies are available (e.g. Montabone et al., 2015). GCM simulations with radiatively active dust will provide an adequate description not only of tides but also of the mean meridional circulation and the diabatic processes due to solar heating of the dust during the day and radiation to space at night. Wind fields generated by the GCM would provide detailed insight in the processes driving the observed variability and could be used to drive Lagrangian particle simulations along realistic trajectories. I believe an interpretation of the observed effects is not adequate without considering Mars GCM simulations.

My final comment concerns the section of total water content. While the approach of calculating total water from ice observations and the saturated vapor pressure at a given temperature has been used before, the attempt of using this approach to determine diurnal variations of water is fatally flawed. The method relies on the existence and detectability of clouds. However, cloud occurrence is heavily influenced by tidal temperature variations (e.g. Lee et al., 2009) so the variation in total water that is claimed in the manuscript (supplementary figure 7) might be just a reflection of changes in temperature due to the tide and subsequent ice formation or sublimation. In addition, due to temperatures being generally higher during dust storms, water might well be present even if not clouds are observed. The claim that the tidal behavior provides a fast transport mechanism from a polar source to high altitudes in mid-latitudes is also questionable at best as it requires mixing of polar air with mid-latitude air, which is not considered in the manuscript. Polar vortices tend to be quite robust dynamic systems in the Martian winter atmosphere and the evaluation of water transport out of the polar region would again at least require the consideration of GCMs that simulate this effect.

In all, I cannot recommend the publication of this article in Nature Communications.

References:

Montabone, L., et al. (2015) Eight-year climatology of dust optical depth on Mars, *Icarus* 251, 65-95, doi: 10.1016/j.icarus.2014.12.034.

Lee, C., et al. (2009) Thermal tides in the Martian Middle Atmosphere as Seen by the Mars Climate Sounder, *J. Geophys. Res.* 114, E03005, doi: 10.1029/2008JE003285.

Reviewer #3 (Remarks to the Author):

Dust Tides and Fast Meridional Motions in the Martian Atmosphere During Major Dust Storms, by Zhaopeng Wu, Tao Li, Xi Zhang and Jun Cui.

This paper identifies significant diurnal variability in the dust distribution of the Mars atmosphere during periods of significant dust lifting. The diurnal variation is particularly prominent at high latitudes in the summer (southern) hemisphere in a layer of atmosphere roughly 10-40 km above the surface. The authors convincingly demonstrate that this dust behavior can be explained by horizontal advection provided by the (quasi) horizontal, planetary-scale wind circulation associated

with the sun-synchronous (migrating) diurnal-period thermal tide. These results are new and have not appeared in the published literature. However, the coupled influence of tides and aerosol have recently attracted interest. In particular, a study was presented at the 9th International Mars conference in Pasadena in July 2019 [Kleinbohl et al. [2019]] and the 2-page abstract is available on line.

The manuscript is generally logically laid out and is well-illustrated. I suspect that the authors need to provide further background on thermal tides for the subject to be more clear to a broad audience. The spacecraft observations on which this work is based are obtained in a sun-synchronous orbit, so that the local times of observation are fixed. This provides adequate coverage to constrain the temperature response by the diurnal period, sun-synchronous (migrating) zonal wave 1 thermal tide, which is expected to be dominant during dusty periods when aerosol heating is the dominant forcing. Tide theory, with reasonable approximations, is invoked to derive the associated wind field, which is used in a dust advection calculation to obtain dust field evolution similar to that observed. I think this approach is adequate for the task, though a better approach is almost certainly to make use of Mars global circulation models which more realistically treat the forcing of thermal tides and self-consistently represent the winds and tracer transport.

I see no reason why the paper can't be published, but would recommend some attention to the comments listed below.

Minor Comments:

General comment: I think it would be preferable to substitute "rapid" for "fast" in many locations through the manuscript.

Line 22: It is stated that the significance of rapid transport of dust and water is often underestimated in Martian climate models due to the lack of diurnal data. Of course, this transport is explicitly represented in models. It would be more accurate to say that detailed comparisons of the simulated diurnal variations are limited by relatively sparse observations. Most spacecraft observations are obtained from sun-synchronous orbit, typically viewing only 2 local times.

The abstract discusses water transport, but the manuscript barely deals with this issue.

Line 27: The concept of "zonally circled dust fronts" is poorly phrased.

Line 49: How are major storms defined? Most of the events from MY28 to MY34 would be considered significant regional storms, occurring in the the so-called A and C seasons, according to Kass et al. [2016].

Line 54: I believe this means that there is little diurnal variability in the dust field during relatively clear conditions.

Line 66: Need to define what a "A regional storm" is. Reference to paper by Kass et al. [2016]. This issue is somewhat addressed in the section starting at Line 211.

Line 69: Is there a reference on previous expectations of diurnal variability of dust height?

Line 79: The A- (and C) season storms actually originate as flushing storms that cross the equator into the southern hemisphere. The dust column opacity and mid-level (50 Pa) temperature responses of these events span a broad range of latitudes. However, it is apparent that the impact on dust height diurnal variability is largely confined to high southern latitudes.

Line 81: Why choose this arbitrary time reference (days from Ls 196)? I think it is preferable to simply use Ls

Line 99; the observed phase of 1800 LT is in good agreement with modeling: see figure 8 in Banfield et al. 2003;

Line 108: "...while those in the high latitudes maintains at low levels". This is difficult to follow. I imagine the authors mean that dust lifting and convection are not present at high latitudes?

Line 113: Change "seasonally varied" to seasonally varying.

Line 140: Of course, the tide wind circulation could include vertical motion. However this is somewhat limited at higher latitudes, where the tide is quasi nondivergent, essentially acting as a vertically-trapped response to diurnal forcing.

Line 148: Maximum polarward advection of dust at ~1800 LT at high latitudes and altitudes is reasonable, as the phase of the tide temperature response is largely centered at ~1800 LT throughout the depth of the dust forcing region. This is not necessarily the case in the tropics, where the diurnal tide typically has a vertically-propagating character with a finite vertical wavelength. This is significantly altered during periods of strong dust forcing, and would not be expected to be properly represented in this idealized tide model.

Line 143: You are trying to say that you are constructing a diurnally-resolved horizontal wind field. The tide response at mid to high latitudes has relatively uniform phase over a broad range of altitudes so that it is reasonable to formulate a fairly representative wind field that can plausibly account for the observed dust advection. Note that the tide fields are more complicated at tropical (typically equatorward of 30 degrees), as the tide is a vertically propagating gravity wave, and is strongly influenced by the zonal mean winds.

There is a need to distinguish between low-level and upper level tide winds, which are out of phase with each other. This is illustrated in Wilson et al. 2008; (conference abstract) and Wilson, 2012 (conference abstract, figure 4). The focus here is on the behavior of dust at "high" altitudes.

Line 170: Change "form" to "from". That tides can induce a net enhanced meridional transport of dust is described in Wilson [1997].

Line 161: I'm uncomfortable with this discussion of water transport. MCS only provides observations of water ice cloud opacity profiles, so discussion of total water content requires assumptions about water vapor, which may be related to temperatures. This is mentioned on line 391-395.

Line 176. The discussion relates to supplementary figure 7, which focuses on MY33 and MY34. Yet the text refers to MY28. Line 401 indicates that the data quality for MY28 is suspect.

Line 189: I think you mean "along-track" instead of "off-track".

Line 193: A word of caution is in order for handling the cross-track data. At low latitudes, the time coverage is more like 0300 +/- 0130 and 1500 +/- 0130. While this provides 6 local time observations/sol, these are very unevenly spaced and the error estimates on the semidiurnal harmonic are quite large. It is adequate for the purposes of this paper to state that the diurnal harmonic can be estimated with good confidence. At high latitudes, the diurnal harmonic of temperature is clearly dominant.. and this is also evident in the dust fields shown.

Line 207: High dust opacity limits the ability of MCS to make retrievals of dust and temperature. However, the dust height can be monitored quite well, and this is described in Kleinbohl et al. [2019] for the MY34 global dust storm.

Line 243: I don't understand the re-binning of multiple local times together (3 hours). Judging from the caption for Figure 3, this evidently amounts to binning at 8x/sol. I don't see the need for this. The figure is effectively showing am (~3am) and afternoon (3pm) opacities.

Line 259: The least squares fitting described is very ill-posed for the semidiurnal and higher harmonics. It can produce satisfactory results for the diurnal period tide. The semidiurnal tide and the stationary wave can be significantly aliased.

Line 285: It is stated that the wind measurement in the Mars atmosphere is poor. In fact, they are effectively non-existent.

Line 290: Change "ideally condition" to "ideal conditions..." These are basically the assumptions of classical tide theory.

Line 391: It is conceded that dust opacity far exceeds water ice cloud opacity during storm conditions. Therefore very little is said about the transport of water. I recommend that this section be eliminated.

Figure 1: I was not sure what was meant by "the altitude from MCS is zonally-averaged from each latitude".

Figure 2: The caption refers to supplementary figure 4, but this ought to be SF 5. Change "The entire period starts from ... " to "The period runs from Ls=195 to 252." Actually, it is preferable to use Ls for the x-axis rather than the Martian day number.

Figure 4: You might point out that the phases of U and V are consistent with those shown in supplementary figure 6. The figure would be simpler if the x-axis were simply labeled with local time. Of course, longitude and local time are equivalent for the migrating tide. The wind fields in panels c and d should be simply phase-shifted versions of each other, as they appear to be. I expect that the dust field at the two times 12 hours apart would also be phase shifted versions of each other, since possible vertical motion is not being accounted for.

Supplementary Figure 2. The caption ought to spell out DH = Dust Height. Same for SF 4.

Supplementary Figure 5. Remove "zoom-in" from the caption and simply state that this is the vertical and meridional structure of high latitude summer hemisphere dust opacity. I also suggest

replacing “...during A regional dust storm...” with “...during the A-season regional dust storm...” It is preferable to use Ls for the x-axis rather than the Martian day number. This applies to Figure 2 as well.

Given the rather restricted local-time coverage, I believe it would be preferable to simply show the “am” and “pm” zonal mean dust fields, as is done in the Kleinbohl et al. [2019] abstract. Such a figure clearly captures the significant diurnal difference in the meridional and vertical extent of the high-latitude dust field.

References:

Banfield, D., B.J. Conrath, M.D. Smith, P.R. Christensen, and R.J. Wilson, 2003: Forced waves in the martian atmosphere from MGS TES nadir data, *Icarus*, 161, 319-345.

Barnes, J. et al., 2017: The General Circulation. *The Atmosphere and Climate of Mars*.

Kleinböhl, A., A. Spiga, D. M. Kass, J. H. Shirley, E. Millour, L. Montabone, and F. Forget (2019). Diurnal variations of dust from Mars Climate Sounder Observations during the global dust event in Mars Year 34. <https://www.hou.usra.edu/meetings/ninthmars2019/pdf/6146.pdf>

Wilson, R.J., and M.I. Richardson, 2000: The Martian Atmosphere During the Viking Mission, 1: Infrared Measurements of Atmospheric Temperatures Revisited. *Icarus*, 145, 555-579.

Wilson, R.J., R.M. Haberle, J. Noble, A.F.C. Bridger, J. Schaeffer, J.R. Barnes, and B.A. Cantor, Simulation of the 2001 planet-encircling dust storm with the NASA/NOAA Mars general circulation model, Mars atmosphere: modeling and observations workshop, Williamsburg, VA, November, 2008. <http://www.lpi.usra.edu/meetings/modeling2008/pdf/9023.pdf>

Wilson, R.J. Martian dust storms, thermal tides and the Hadley circulation, Abstract 8069, Comparative Climatology of Terrestrial Planets, Boulder, CO, June 2012. <http://www.lpi.usra.edu/meetings/climatology2012/pdf/8069.pdf>

Responses to reviewers (in blue)

We thank Dr. Nicholas Heavens and the other two anonymous referees for the constructive comments and suggestions. We have substantially modified the manuscript and included more supplementary figures accordingly. The general overview of revisions is as below:

1. An introduction section has been added prior to the results and discussion sections to comply with the editorial format and policies of Nature Communications. An overview of the background such as on thermal tides is included as suggested by reviewer #3 to be clearer to broad audiences.
2. More descriptions and discussions on the treatment of the MCS (Mars Climate Sounder) data are added to validate our analysis method using the 3-hour local time averages and evaluate the potential biases mentioned by reviewer #2.
3. We have addressed the major criticism of reviewer #2 by validating our methodology using the GCM-based simulations. GCM Simulation results including wind fields and aerosols transport from Mars Climate Database version 5.3 (MCD 5.3) is added to compare with the derived wind results in this work to evaluate and validate our methodology. The comparison shows good agreement between our simple linear theory and GCM simulation.
4. As suggested by all the three reviewers, the discussion of total water content directly calculated from observations has been eliminated as its method is potentially debatable. Instead, simulation results from MCD 5.3 and discussions of water vapor transport are added to support our proposed mechanism of rapid transport of moist air from summer pole to lower latitude. Our previous implication in the work on the water escape is still valid and important for the Martian climate.
5. Improvement of the figures and phrases has been made according to suggestions of the reviewers. We have also greatly improved the writing in the revised manuscript.

The detailed modifications are highlighted below in blue in the replies.

Reviewers' comments:

Reviewer #1 (Remarks to the Author):

In this manuscript, Wu et al. demonstrate that during major dust storms on Mars, a volume of high dust concentrations (or "dust front") migrates westward along with the Sun in line with the main westward-migrating diurnal tide. Mirroring this dust front, less dusty air migrates westward opposite to the Sun on the nightside. The conclusion drawn by the authors is that the dust front originates from rapid meridional transport at 10-40 km altitude (winds of almost 90 m/s) from dusty air in the mid-latitudes. In that case, less dusty and presumably moist air from the polar cap is likely being rapidly transported on the nightside to the mid-latitudes. Such a water supply, if entrained by dusty deep convection at low-mid latitudes, will effectively bring water from the polar cap to high in the middle atmosphere, where it can be photodissociated and enhance the supply of hydrogen that can escape from Mars's atmosphere.

The manuscript is awkwardly written and the figures are not always as impactful as they should be. That said, the manuscript is well-organized, well-argued, and methodologically expansive. Its claims should be of major interest to the Mars atmospheric community and anyone who studies the long-term evolution of Mars's climate. I therefore think the manuscript should be suitable for publication in Nature Communications after revision.

At this point, I should note that the novelty of the claims may not appear so novel by the time it is published. Armin Kleinböhl and Aymeric Spiga in particular have been studying this phenomenon within the 2018 global dust storm: <http://adsabs.harvard.edu/abs/2018AGUFM.P43J3872K> and <https://www.hou.usra.edu/meetings/ninthmars2019/pdf/6146.pdf>. And it is my understanding that a relevant manuscript is under review.

Reply: We have greatly improved the writing in the revised manuscript. Thanks for reminding us that other research groups are working on similar topics. We also wish to hear back from the editor and reviewers as soon as possible.

However, this manuscript surpasses any competing manuscript in terms of the scope of the dust storms analyzed and the elegance of the demonstration of the association between the DW1 tidal mode and the dust front.

As it stands, the manuscript makes three major claims:

1. There are rapidly migrating dust fronts during major dust storms on Mars.
2. These dust fronts are driven by the westward migrating diurnal thermal tide.
3. These fronts will transport moist air from the summer pole to lower latitude.

Reply: Thanks for the recognitions. As has been summarized above by Dr. Heavens, a relevant manuscript is currently under review by another journal, but that work only focuses on a specific dust storm, i.e., the global dust storm in Mars Year (MY) 34. On the other hand, our work has much more broader time coverage and much larger impact than the other work. First, we discovered and discussed rapidly migrating dust fronts in all major dust storms in total 7 Mars Years, which makes the dust tide phenomenon more general and significant. Second, our methods can provide a more intuitive illustration to broad audiences, because we not only used numerical simulations but also a direct demonstration from the data on the relationship between the diurnal tide and the dust front. Finally, we extended the dust phenomenon to a broader scope associated with water vapor escape, which is very important for Mars climate evolution. With the help of GCM simulations, we also proposed a mechanism on transporting moist air from summer pole to lower latitude and then potentially contribute to water escape by deep convections during the southern-summer-season global dust storms, such as that observed in MY 28. We want to point out that, this mechanism, however, is not applicable to the global dust storm in MY 34 (because it is in the southern spring season). Therefore, we do not think the other work under review by another journal is able to address this issue.

The first claim is supported by Figure 1 (along with Supplementary Figure 1), but Supplementary Figure 1's format is different enough from Figure 1 to undermine the argument for the general reader. The Supplementary Figure needs to show examples of what 10 days of diurnal change in dust opacity would look like prior to commencement of the dust storms in Figure 1.

Reply: Thank you for your suggestion. In our new Supplementary Fig. 1, we have plotted 10 days of diurnal change in dust opacity before or after the periods of major dust storms shown in Fig. 1. We cannot show the time evolutions prior to commencement of all dust storms in Fig. 1 because some of them are not within the cross-track observation periods as shown in the new Supplementary Fig. 4. Instead, we show the time series after the dust storms for those cases. This should support the argument for the general reader as well.

The second claim is elegantly supported by Figure 2.

The third claim is partly supported by Figure 4 and Supplementary Figure 5 but requires a great deal of unreferenced speculative language at lines 168-171 to make it across the line. The manuscript does not make it clear how exactly water vapor would escape from the tidal trap. Discussion of the third claim also needs to emphasize that the mechanism only works if transport is from the summer pole (always the southern pole in the cases examined by the manuscript). Otherwise, nightside transport would be from air much drier than that at the mid-latitudes. Typical Martian dewpoints are 170-190 K, while winter pole cap temperatures are 140 K.

The recommendation I would make about the third claim is to emphasize that any sufficiently strong circulation at lower latitudes that is asymmetric with the westward-propagating diurnal tide in the southern mid-latitudes will result in net meridional transport. One such circulation (a very slow-moving wave) is suggested by Clancy et al. (2010, doi: 10.1016/j.icarus.2009.10.011). I am loathe to recommend my own work, but Fig. 14a-c,e-g in Heavens et al. (2019,

10.1175/JAS-D-19-0042.1) shows likely examples of deep convective mesoscale circulations in the southern mid-latitudes in the early stages of major dust storms.

Reply: Thanks for the valuable comments and suggestions. We have added examples of the simulated meridional transports of water vapor in different seasons from MCD 5.3 in Supplementary Fig. 10 and the Discussion section to emphasize the applicability of the proposed mechanism. The new results clearly show that nightside transport of water from higher latitudes to lower latitudes only works in southern summer season when water vapor is sublimated from polar cap water ice. And we have also added more discussions and citations to make it clear how water vapor would escape from the tidal trap.

I do not want to spend any time copyediting the manuscript, which is a matter, I think, for the authors and the editors of the journal. But I found it easier to understand the manuscript by rewriting the abstract in this way:

"Mars's atmosphere is strongly affected by the spatial and temporal variability of airborne dust.1-4. Observations have shown interannual, interseasonal, and diurnal variability in the dust distribution 5-9. However, the diurnal variation – global dust change within a sol (Martian day) – is still poorly understood10,11. Although short-term dynamic processes are crucial to rapidly transporting dust and water to higher altitudes12,13, their significance is often underestimated in Martian climate models1 due to the lack of diurnal data. Here we report the discovery of ubiquitous, strong diurnal tides of dust in the southern hemisphere of Mars. Using data with a relatively high local time resolution from the Mars Climate Sounder, we found that zonally circled dust fronts of 10-40 km high slosh back and forth in a wide latitudinal range up to 40° (~2300 km) within one sol during major dust storms. We show that these dust fronts are primarily driven by the westward-propagating diurnal thermal tide. Tidal transport of dust in this way would rapidly transport heat and constituents meridionally, allowing moist air near the summer pole to be rapidly transported to lower latitudes at night, where it then can be lifted by deep convection during the day and contribute to hydrogen escape from Mars during global dust storms 12."

Reply: Thank you very much for the suggested rewriting. We have updated our abstract with modifications.

Minor comments:

Abstract: I do not think "zonally circled" works well to describe the dust fronts. They are actually oriented in the meridional direction but travel zonally.

Reply: We have changed it to "zonally distributed dust fronts" in the revised manuscript.

Line 35: "is only" rather than "only possesses"

Reply: In the revised paper we have deleted this sentence but added equivalent descriptions in the introduction section.

Lines 52-53: This claim about processes is unsupported.

Reply: Thanks for pointing this out. We now discuss the possible mechanisms later in the paper. We have modified this sentence from “The main dust filling-up and scavenging process occurs between 10 and 100 Pa” to “The diurnal variation of dust mainly occurs between 10 and 100 Pa”.

Lines 68-69: Perhaps rephrase to "where major dust storms are known to enhance diurnal thermal tides"

Reply: Modified as suggested.

Line 70: Perhaps rephrase to: "Outside major dust storms, the DH day-night variability of up to 10 km is not uncommon."

Reply: Modified as suggested.

Line 70: "Previously expected" by whom?

Reply: This sentence has been rephrased to “Outside major dust storms, the DH day-night variability of up to 10 km is not uncommon.”

Lines 87-90: Particularly awkward.

Reply: We have rephrased this sentence to “The migrating semidiurnal tide and stationary planetary waves were previously found common and strong in Martian atmosphere, but they are much weaker than DW1 by up to one order of magnitude during major dust storms.”

Line 99: "has a dramatic shift"

Reply: Modified as suggested.

Line 105: "dramatic" rather than "dramatical"

Reply: Modified as suggested and so did the other places.

Line 136: "Rapid meridional motion of the dust front implies..."

Reply: Modified as suggested.

Lines 172-173: As noted in the Methods, the water content proxy is limited to 30 km altitude and non-dusty atmosphere, primarily tracer of water getting high in the middle atmosphere; it is a bit of a stretch to use it to diagnose transport over the 10-40 km altitude range mentioned in the abstract.

Reply: Thanks for the comment. The other two reviewers also raised the same concern and suggested to eliminate this part. Therefore, we have replaced the discussion on the total water content with the simulated water vapor results from MCD 5.3 which shows the same rapid meridional motion and therefore does not alter our conclusions.

Supplementary Figure 6: The layout is overly confusing. An improvement would be to put the e-h colorbars and labels at the bottom of the figure.

Reply: Modified as suggested.

Nicholas Heavens

Reviewer #2 (Remarks to the Author):

Review of manuscript “Dust Tides and Fast Meridional Motions in the Martian Atmosphere During Major Dust Storms” submitted to Nature Communications

This manuscript presents results of analyses of the spatial and temporal variabilities of dust in the Martian atmosphere during global and large regional dust storms. Analyses are based on profile measurements of temperature and dust by the Mars Climate Sounder (MCS) remote sensing instrument. The manuscript illustrates variations in the zonally averaged dust between daytime and nighttime, with daytime dust extending much farther into the southern high latitudes than nighttime dust during major dust storms. The authors use so called cross-track measurements by MCS to extend the local time coverage of their analysis. The authors fit tidal expressions to the temperature and dust data, and use fundamental equations to calculate zonal gradient winds and tidal winds. With these idealized wind fields a Lagrangian particle analysis is performed, in which virtual particles are initialized and tracked as they are being moved by the zonal and tidal winds. The analysis suggests that the diurnal variation of the dust is created by wind fields that are altered according to the westward migrating diurnal tide. In addition, the manuscript presents calculations of total water content based on water ice opacities observed by MCS. It argues for a meridional motion of the total water content during a Martian day and suggests a fast transport mechanism from a polar source to mid-latitudes at night and lofting to higher altitudes during the day, which may lead to enhanced hydrogen escape.

The manuscript describes features in the Martian atmosphere that are likely to be interesting to scientists specialized in the field of atmospheric dynamics. The variability of aerosols, in particular on sub-diurnal time scales, is an area that is understudied in the Martian atmosphere. The manuscript provides a credible description of the dust variability observed by the Mars Climate Sounder. However, I do have significant concerns about some of the methodologies applied to interpret the observations, which I will describe in more detail below. Some methodologies are not state-of-the-art while at least one seems to be fatally flawed, leading to conclusions that are unsupported and possibly wrong. Due to these concerns I cannot recommend the manuscript for publication by Nature Communications.

Reply: Thanks for the recognitions and criticisms which have greatly helped us improve this manuscript.

One of my points of criticism concerns the treatment of the MCS cross-track data. MCS cross-track measurements were only obtained in campaigns. While the availability of these measurements is displayed in supplementary figure 3, the detailed characteristics of these data does not seem to be considered adequately. The time difference of cross-track measurements to standard measurements along the orbit track varies significantly with latitude, from +/- 1.5 hours at the equator to over +/- 3 hours at higher latitudes. Several parts of the analysis consider 3-hour local time averages (e.g. figure 3, supplementary figures 5, 7). These somewhat arbitrarily selected time windows are likely to contain mixtures of cross-track and along-track measurements, which obscures the actual local time coverage that goes into the averages, potentially causing biases in

the results. Such biases have not been discussed at all in the manuscript. In addition, some figures show data that contains cross-track measurements next to data that does not (e.g. the dust storm in Mars year 28 does not have any cross-track coverage, while the storms in Mars years 33 and 34 do). This is likely to cause biases in the comparisons, especially when data across a large range of latitudes are considered. Also, zonal averages at the onset of large-scale dust storms (such as shown in figure 1) may not be very meaningful as dust activity and dynamical responses can be quite localized at the onset of dust storms. No consideration of this is provided in the manuscript.

Reply: Thanks for these comments.

There were three figures (Fig. 3, Supplementary Fig. 5 and Fig. 7) in the original manuscript that used the time windows of 3 hours for data binning.

For Fig. 3: In the revised manuscript, to avoid the biases of adjacent local times observed by cross-track strategy, we used the shorter time window of 1 hour (2.5-3.5 a.m. averaged for ~3 a.m. and 2.5-3.5 p.m. averaged for ~3 p.m.) to replot Fig. 3. The new Fig. 3 shows almost the same behavior as in the old figure except there are some missing data in the polar region (higher than 77.5°S). This confirms that the binning scheme does not alter our previous conclusions.

For Supplementary Fig. 7 (original): We have eliminated it because it is related to the total water content. Since this method is potentially debatable, so we have eliminated previous discussions. Instead, in the revised manuscript we added the GCM-based simulation results of water vapor from MCD 5.3 [Millour et al., 2018], as suggested by the reviewer.

For Supplementary Fig. 5 (Supplementary Fig. 6 in the revised manuscript): We agree with the reviewer's comment that our data analysis contains mixtures of cross-track and along-track measurements, which obscures the actual local time coverage that goes into the averages, potentially causing biases in the results. Therefore, we added a new Supplementary Fig. 7 and the related descriptions for evaluation of the potential biases. We think this figure can give a better illustration on where the biases comes from and to what degree it can be. As shown in the new figure, the time window of 3 hours is selected on careful consideration to allow enough data coverages in latitude for the five different local time plots in Supplementary Fig. 6. As a compromise, mixtures of cross-track and along-track measurements are needed in the averages for some local time bins. But the biases are no more than the selected time window and do not alter our previous conclusion on the diurnal variation of the dust front. In the revised version, the potential biases caused by the uneven distributed data point within each bin grid is represented by the horizontal (equal to the local time bin interval) and vertical (according to the latitude bin interval) error bars for reference.

For zonal averages at the onset of large-scale dust storm: We agree with the reviewer's opinion that the onset of dust activity and dynamical responses can be localized. We have mentioned the "onset" term in the old main text just to emphasize the rapid development of the dust storms as a baseline for comparison with the evolutions of the diurnal tide components. We did not discuss it in detail since it is beyond the scope of this work. However, in the revised

manuscript, to avoid misleading the audiences we have added a brief introduction about the onset of the large dust storms in the “major dust storm” section in Method and avoided using the specialized term “onset” in the main text. In addition, the Figures 1c,d,e which the reviewer mentioned to have zonal averages at the “onset” of large-scale dust storms in our old manuscript have been re-plotted and only show zonal averages after the “onset” of the dust storms in the revised manuscript.

Another and more substantial point of criticism concerns the methodology of deriving the wind fields for the simulation of their particle trajectories. Zonal winds are derived from gradient winds and are considered constant over the course of the Martian day, while time-dependent winds are derived from tidal wind equations. While both expressions are valid equations, they have severe limitations when applied to real-world meteorological conditions. Gradient winds are only available over a certain range within the mid-latitudes, and require an assumption of “no motion” (line 277) at a certain pressure level, which is not necessarily true. The tidal winds require “dissipation-less, zero mean wind and no horizontal temperature gradient” (line 292-293). Virtually all these assumptions are not fulfilled. Gradient winds have been assumed as mean winds, which are obviously not zero. Horizontal temperature gradients are significant, certainly along longitude circles but even along latitude circles, e.g. due to stationary waves and non-migrating tides. Finally, the Martian atmosphere has a very short radiative time constant, so it cannot be considered dissipation-less. While the authors acknowledge that diabatic heating is not zero (line 337) they argue that this effect would only influence the vertical wind but would have limited effects on horizontal wind structure. No evidence is provided for this statement. In contrast, the authors show vertical wind in supplementary figure 6 and discuss it in the manuscript (lines 352-362).

The derivation of wind fields from fundamental equations using the aforementioned assumptions cannot be considered state-of-the-art. The way to quantify such processes nowadays is through analyses of simulations from General Circulation Models (GCMs). Modern Mars GCMs are well capable of simulating tidal processes such as the ones dealt with in this manuscript. Typically they will have to be driven with a dust climatology as surface lifting of dust is not simulated well enough but such climatologies are available (e.g. Montabone et al., 2015). GCM simulations with radiatively active dust will provide an adequate description not only of tides but also of the mean meridional circulation and the diabatic processes due to solar heating of the dust during the day and radiation to space at night. Wind fields generated by the GCM would provide detailed insight in the processes driving the observed variability and could be used to drive Lagrangian particle simulations along realistic trajectories. I believe an interpretation of the observed effects is not adequate without considering Mars GCM simulations.

Reply: Thanks for the comments and suggestions. First of all, we agree with the reviewer that the classical theories we used to derive the wind field have more assumptions than the GCMs. The gradient wind theory and classical tidal theory are all linear theories and have explicit solutions, which are convenient for us to be directly applied to the observational temperature data. However, we do not try to do a quantitative, detailed comparison with the actual observation in this work. Instead, using the classical theories, we aimed to achieve a simpler, physically intuitive understanding of the dust tides. In fact, we evaluated the horizontal circulation pattern and found

that it can account for the diurnal variation of the dust front especially in the phase. The classical theories succeed. We also believe those simple, intuitive understanding is important for general readers of this journal. As reviewer #3 pointed out, the classical theory should be “adequate for the task”.

But yes, we also agree that modern Mars GCMs are state-of-the-art way in atmospheric science research and capable of simulating tidal processes. Therefore, a major revision in our resubmission is including the simulated wind pattern from MCD 5.3 which is based on the state-of-the-art LMD GCM (Laboratoire de Météorologie Dynamique General Circulation Model [Forget et al., 1999]). We show, in Supplementary Fig. 9, a good agreement between the derived wind field based on our classical theories and simulated wind field from MCD 5.3. This further validates our approach using the classical theory. In fact, the agreement has provided two potential insights: (1) while numerical simulations have been regarded as the state-of-the-art method for atmospheric research, the classical theories are still useful for qualitative analysis to understand the fundamental physical mechanism. And the assumptions used in the classical theories apparently do not result in large biases from the GCM simulations for our task. (2) The dominant dynamic process for the meridional motions of airborne dust between ~10-100 Pa in the southern hemisphere during major dust storms is westward-propagating migrating diurnal thermal tide. Extreme weather events (such as major dust storms) can be treated as natural control experiments that may intensify one process (such as tidal process) more than the others.

We have no replies to the specific criticisms on the assumptions used in our method except for the vertical wind issue in lines 352-362. The dust has apparent meridional motion but very limited vertical motion in the mid-high latitudes based on Fig. 3 and Supplementary Fig. 6. The vertical wind is never used in the Lagrangian particle analysis or in any related discussion. Therefore, we have eliminated the vertical wind related part in the manuscript. In addition, we have pointed out the limitation of our methodology of deriving the wind fields in the Results and Discussion sections as a reference to broad audiences. Again, a good agreement of our derived wind field with the simulated wind field in the MCD 5.3 provides a validation on the range of application of the classical theories. With those comparison, we hope the reviewer can agree with our approach now.

My final comment concerns the section of total water content. While the approach of calculating total water from ice observations and the saturated vapor pressure at a given temperature has been used before, the attempt of using this approach to determine diurnal variations of water is fatally flawed. The method relies on the existence and detectability of clouds. However, cloud occurrence is heavily influenced by tidal temperature variations (e.g. Lee et al., 2009) so the variation in total water that is claimed in the manuscript (supplementary figure 7) might be just a reflection of changes in temperature due to the tide and subsequent ice formation or sublimation. In addition, due to temperatures being generally higher during dust storms, water might well be present even if not clouds are observed. The claim that the tidal behavior provides a fast transport mechanism from a polar source to high altitudes in mid-latitudes is also questionable at best as it requires mixing of polar air with mid-latitude air, which is not considered in the manuscript. Polar vortices tend to be quite robust dynamic systems in the Martian winter atmosphere and the evaluation of

water transport out of the polar region would again at least require the consideration of GCMs that simulate this effect.

Reply: Thanks for this important comment. We have noticed that the method of calculating total water content is debatable as suggested by all the three reviewers, so we have eliminated the initial discussions. Following the reviewer's comments, in the revised manuscript we added the simulation results of water vapor from the MCD 5.3 instead. The GCM results clearly show rapid meridional motion as we proposed. We have also added more discussions including those suggestions by reviewer #1 and #3 for the possible mechanisms of air mixing at mid-latitudes.

In all, I cannot recommend the publication of this article in Nature Communications.

Since the paper has been substantially revised, hopefully the current form is acceptable now.

References:

Montabone, L., et al. (2015) Eight-year climatology of dust optical depth on Mars, *Icarus* 251, 65-95, doi: 10.1016/j.icarus.2014.12.034.

Lee, C., et al. (2009) Thermal tides in the Martian Middle Atmosphere as Seen by the Mars Climate Sounder, *J. Geophys. Res.* 114, E03005, doi: 10.1029/2008JE003285.

Reviewer #3 (Remarks to the Author):

Dust Tides and Fast Meridional Motions in the Martian Atmosphere During Major Dust Storms, by Zhaopeng Wu, Tao Li, Xi Zhang and Jun Cui.

This paper identifies significant diurnal variability in the dust distribution of the Mars atmosphere during periods of significant dust lifting. The diurnal variation is particularly prominent at high latitudes in the summer (southern) hemisphere in a layer of atmosphere roughly 10-40 km above the surface. The authors convincingly demonstrate that this dust behavior can be explained by horizontal advection provided by the (quasi) horizontal, planetary-scale wind circulation associated with the sun-synchronous (migrating) diurnal-period thermal tide. These results are new and have not appeared in the published literature. However, the coupled influence of tides and aerosol have recently attracted interest. In particular, a study was presented at the 9th International Mars conference in Pasadena in July 2019 [Kleinbohl et al. [2019]] and the 2-page abstract is available on line.

The manuscript is generally logically laid out and is well-illustrated. I suspect that the authors need to provide further background on thermal tides for the subject to be more clear to a broad audience. The spacecraft observations on which this work is based are obtained in a sun-synchronous orbit, so that the local times of observation are fixed. This provides adequate coverage to constrain the temperature response by the diurnal period, sun-synchronous (migrating) zonal wave 1 thermal tide, which is expected to be dominant during dusty periods when aerosol heating is the dominant forcing. Tide theory, with reasonable approximations, is invoked to derive the associated wind field, which is used in a dust advection calculation to obtain dust field evolution similar to that observed. I think this approach is adequate for the task, though a better approach is almost certainly to make use of Mars global circulation models which more realistically treat the forcing of thermal tides and self-consistently represent the winds and tracer transport.

I see no reason why the paper cant be published, but would recommend some attention to the comments listed below.

Reply: Thanks for the recognitions.

Minor Comments:

General comment: I think it would be preferable to substitute “rapid” for “fast” in many locations through the manuscript.

Reply: Modified as suggested.

Line 22: It is stated that the significance of rapid transport of dust and water is often underestimated in Martian climate models due to the lack of diurnal data. Of course, this transport is explicitly represented in models. It would be more accurate to say that detailed comparisons of the simulated diurnal variations are limited by relatively sparse observations. Most spacecraft observations are obtained from sun-synchronous orbit, typically viewing only 2 local times.

Reply: Thanks for pointing this out. This sentence has been changed to “Although short-term dynamic processes are crucial to rapidly transporting dust and water to higher altitudes, detailed comparisons of the simulated diurnal variations are limited by relatively sparse observations.” And based on the comparison between our results and simulated diurnal variations of the water vapor in the MCD 5.3, the meridional transport is indeed explicitly represented in the Martian climate models.

The abstract discusses water transport, but the manuscript barely deals with this issue.

Reply: We have added the simulation results and discussion of water vapor transport from MCD 5.3 in the discussion section to support this argument.

Line 27: The concept of “zonally circled dust fronts” is poorly phrased.

Reply: We have changed it to “zonally distributed dust fronts” in the revised manuscript.

Line 49: How are major storms defined? Most of the events from MY28 to MY34 would be considered significant regional storms, occurring in the so-called A and C seasons, according to Kass et al. [2016].

Reply: Thanks for the suggestion. The definition of the major dust storms has been added in the “Major dust storms” section of Method.

Line 54: I believe this means that there is little diurnal variability in the dust field during relatively clear conditions.

Reply: Yes. To make it clearer we have changed it to “On the contrary, the variation of the dust opacity before and after the periods of major dust storms is much weaker within a sol (Supplementary Fig. 1)” Besides, we have replaced the old supplementary Fig. 1 with examples of 10 days of diurnal change in dust opacity before and after the periods of major dust storms (in the dust field during relatively clear conditions).

Line 66: Need to define what a “A regional storm” is. Reference to paper by Kass et al. [2016]. This issue is somewhat addressed in the section starting at Line 211.

Reply: Thanks for the suggestion. We have added the definitions of A, B and C regional storms in the “Major dust storms” section of Method.

Line 69: Is there a reference on previous expectations of diurnal variability of dust height?

Reply: As far as we know there is no previous discussions on the globally diurnal variability of dust height similar as we define in this paper. So, this phrase (stronger than previously expected) is not proper here. Also, as reviewer #1 suggested, we have changed this sentence to “Outside major dust storms, the DH day-night variability of up to 10 km is not uncommon”.

Line 79: The A- (and C) season storms actually originate as flushing storms that cross the equator into the southern hemisphere. The dust column opacity and mid-level (50 Pa) temperature responses of these events span a broad range of latitudes. However, it is apparent that the impact on dust height diurnal variability is largely confined to high southern latitudes.

Reply: Thanks for this comment. We have noticed that the old phrase “the A regional dust storm of the southern mid-high latitudes in MY 33” is misleading. In the revised manuscript we changed the sentence to “We selected the A regional dust storm of MY 33 to investigate the evolution of the dust diurnal variation in the southern mid-to-high latitudes.”

Line 81: Why choose this arbitrary time reference (days from Ls 196)? I think it is preferable to simply use Ls

Reply: Modified as suggested. This sentence has been changed to “The duration of this major storm is approximately ~30 sols (Ls= $\sim 214^\circ$ to $\sim 233^\circ$).”

Line 99; the observed phase of 1800 LT is in good agreement with modeling: see figure 8 in Banfield et al. 2003;

Reply: Thanks for this comment. This is a good validation for our results and we have added the citation in the revised manuscript.

Line 108: “...while those in the high latitudes maintains at low levels”. This is difficult to follow. I imagine the authors mean that dust lifting and convection are not present at high latitudes?

Reply: Yes, that was what we tried to say, and it was a bit awkward. We changed the sentence to “The dust abundance and height at low-to-mid latitudes are enhanced by dust lifting and deep convection, while those at high latitudes are not.” in the revised manuscript.

Line 113: Change “seasonally varied” to seasonally varying.

Reply: Modified as suggested. Thanks.

Line 140: Of course, the tide wind circulation could include vertical motion. However this is somewhat limited at higher latitudes, where the tide is quasi nondivergent, essentially acting as a vertically-trapped response to diurnal forcing.

Reply: Thanks for the comment. As shown in Fig. 3 and Supplementary Fig. 6, the dust has apparent meridional motion but very limited vertical motion at high latitudes. That’s why we only considered horizontal motions in this work. Since the vertical wind is never used in the Lagrangian particle analysis or in any related discussion, we have eliminated the vertical wind related part in the revised manuscript.

Line 148: Maximum poleward advection of dust at ~1800 LT at high latitudes and altitudes is reasonable, as the phase of the tide temperature response is largely centered at ~1800 LT throughout the depth of the dust forcing region. This is not necessarily the case in the tropics, where the diurnal tide typically has a vertically-propagating character with a finite vertical wavelength. This is significantly altered during periods of strong dust forcing, and would not be expected to be properly represented in this idealized tide model.

Line 143: You are trying to say that you are constructing a diurnally-resolved horizontal wind field. The tide response at mid to high latitudes has relatively uniform phase over a broad range of altitudes so that it is reasonable to formulate a fairly representative wind field that can plausibly account for the observed dust advection. Note that the tide fields are more complicated at tropical

(typically equatorward of 30 degrees), as the tide is a vertically propagating gravity wave, and is strongly influenced by the zonal mean winds.

Reply: Thanks for the comments of lines 143 and 148. Yes, we do realize and see from our results that the vertical wavelength at mid to high latitudes is large (relatively uniform in phase change with height) between 100 and 10 Pa indicating a vertically-trapped mode, while in the low latitudes the vertical wavelength is ~30 km when dust amount is low and the phase change becomes complicated during dust storms which may be influenced by the zonal mean winds as you suggested. The comparison between the derived and simulated wind fields (Supplementary Fig. 9) also shows large biases at tropics as well. We have also added these discussions and cautions to the Results section and “Tidal wind” section in Method. Fortunately, our focus is mainly on the mid-to-high latitudes, so this issue does not affect the conclusion.

There is a need to distinguish between low-level and upper level tide winds, which are out of phase with each other. This is illustrated in Wilson et al. 2008; (conference abstract) and Wilson, 2012 (conference abstract, figure 4). The focus here is on the behavior of dust at “high” altitudes.

Reply: Thanks for this important suggestion. We have added the descriptions in the Discussion section in the revised manuscript.

Line 170: Change “form” to “from”. That tides can induce a net enhanced meridional transport of dust is described in Wilson [1997].

Reply: Thanks for this valuable suggestion. We have added it into the Discussion section.

Line 161: I’m uncomfortable with this discussion of water transport. MCS only provides observations of water ice cloud opacity profiles, so discussion of total water content requires assumptions about water vapor, which may be related to temperatures. This is mentioned on line 391-395.

Reply: Thanks for this comment. We have noticed that this method is potentially debatable as suggested by all the three reviewers, so we have eliminated the discussions of water from MCS. In the revised manuscript we add the simulation results of water vapor instead.

Line 176. The discussion relates to supplementary figure 7, which focuses on MY33 and MY34. Yet the text refers to MY28. Line 401 indicates that the data quality for MY28 is suspect.

Reply: Thanks for this comment. Since the application of this method to evaluate the diurnal variation of total water content is debatable, this line has been eliminated in the revised paper.

Line 189: I think you mean “along-track” instead of “off-track”.

Reply: Thanks for pointing this out. We have changed it to “along-track”.

Line 193: A word of caution is in order for handling the cross-track data. At low latitudes, the time coverage is more like 0300 +/- 0130 and 1500 +/- 0130. While this provides 6 local time observations/sol, these are very unevenly spaced and the error estimates on the semidiurnal harmonic are quite large. It is adequate for the purposes of this paper to state that the diurnal harmonic can be estimated with good confidence. At high latitudes, the diurnal harmonic of temperature is clearly dominant. and this is also evident in the dust fields shown.

Reply: Thanks for this reminds. We noticed it as well when applying this method to higher order harmonics especially during dust storms when the higher order harmonics are weakened [Guzewich et al., 2014]. The diurnal harmonics both in temperature and dust fields especially at mid-high latitudes are valid with modest uncertainties as shown in Fig. 2. In the revised manuscript, we have added this caution to the “MCS dataset” section in Method.

Line 207: High dust opacity limits the ability of MCS to make retrievals of dust and temperature. However, the dust height can be monitored quite well, and this is described in Kleinbohl et al. [2019] for the MY34 global dust storm.

Reply: Yes, we agree with this point. The dust height (usually at altitude of ~ 10 Pa during global dust storms as shown in Fig.3) can be monitored because retrievals at high altitudes (pressures below 20 Pa) are still available during global dust storms. And we had shown the day-night variability of dust heights during all major dust storms including the MY34 global dust storm in Supplementary Fig.2 (Supplementary Fig.3 in the revised version). However, retrievals at lower altitudes (pressures above 20 Pa) are limited due to the high dust opacity as shown in Fig.3 during global dust storms. Figure 1 in Kleinbohl et al. [2019] can show retrievals at these lower altitudes because it includes data of longer time (5° Ls) and more importantly the data are binned separately for day and night which for each part (day or night) it includes all available local times (including the in-track and cross-track observations). In Fig.3, we only shows the results of 3 a.m. and 3 p.m. with a time window of only 1 hour. We can see less retrievals are obtained than that during regional dust storms (Fig.3 a-c). However, many analysis procedures in our work (e.g. nonlinear least squares fitting for Fig. 2b,d, tidal wind derivations for Fig.4 and detailed dust structures at different local times in Supplementary Fig. 6) require good data coverages in local time, longitude and altitude, which are not the case during the global dust storm in MY34. This explains why we select the A regional dust storm in MY33 for the necessary analysis in this work (partly, the other reason concerning the performance of Lagrangian particle analysis is addressed below in the reply to the comment of Fig. 4).

Line 243: I don't understand the re-binning of multiple local times together (3 hours). Judging from the caption for Figure 3, this evidently amounts to binning at $8x/sol$. I don't see the need for this. The figure is effectively showing am ($\sim 3am$) and afternoon (3pm) opacities.

Reply: Thanks for this comment. Yes, Fig. 3 in the main text only shows the variations between 3am and 3pm. And reviewer #2 also raised the concern that the time window of 3 hours may cause biases for comparisons as she/he said “some figures (Fig. 3) show data that contains cross-track measurements next to data that does not (e.g. the dust storm in Mars year 28 does not have any cross-track coverage, while the storms in Mars years 33 and 34 do). This is likely to cause biases in the comparisons, especially when data across a large range of latitudes are considered.” Therefore, to avoid the biases of adjacent local times observed by cross-track strategy, we used the shorter time window of 1 hour (2.5-3.5 a.m. averaged for ~ 3 a.m. and 2.5-3.5 p.m. averaged for ~ 3 p.m.) to replot this figure in the revised manuscript. The new Fig. 3 looks very similar to the old one except with some missing data in the polar region (higher than $77.5^\circ S$). This shows that the new binning scheme does not alter our previous conclusions.

Line 259: The least squares fitting described is very ill-posed for the semidiurnal and higher harmonics. It can produce satisfactory results for the diurnal period tide. The semidiurnal tide and the stationary wave can be significantly aliased.

Reply: Thanks for this comment. The reason we evaluated the semidiurnal tide and the stationary wave was to make sure that the migrating diurnal tide is the dominant wave during dust storms.

Although previous work has suggested this point [Guzewich et al., 2014], we would like to rule out exceptions using new observations. But this evaluation only focuses on mid to high latitudes where the local times are relatively evenly spaced and then the error estimates for the fitting are modest. In the revised manuscript we add more discussions and cautions for use of the cross-track data and the fitting procedure to the “MCS dataset” and “Tidal component fitting” sections in Method.

Line 285: It is stated that the wind measurement in the Mars atmosphere is poor. In fact, they are effectively non-existent.

Reply: Thanks for pointing this out. We have changed “poor” to “non-existent”.

Line 290: Change “ideally condition” to “ideal conditions...” These are basically the assumptions of classical tide theory.

Reply: Modified as suggested.

Line 391: It is conceded that dust opacity far exceeds water ice cloud opacity during storm conditions. Therefore very little is said about the transport of water. I recommend that this section be eliminated.

Reply: Thanks for the suggestion. We have noticed the limitation of this method. This section has been eliminated.

Figure 1: I was not sure what was meant by “the altitude from MCS is zonally-averaged from each latitude”. (Actually it was “for each latitude”, not “from each latitude” in the manuscript)

Reply: MCS retrievals we used in this work are provided on 105 vertical pressure levels. The corresponding altitude data (with uncertainty of 1 km) which is obtained based on the geometric pointing of the instrument is also gridded on these 105 pressure levels [Heavens et al., 2014] (we have introduced this in the Method section). Since we are trying to talk about the dynamic relation between thermal tides and aerosol transport in this paper, we use pressure level coordinate for most descriptions and discussions in this paper. Therefore, the zonal average procedure for dust opacity in Fig. 1 is also performed in pressure level coordinate. But for certain latitude and certain pressure level, the altitude data from MCS in different longitudes are not the same due to topography. Therefore, we zonal average the altitudes for each latitude and pressure level to get an approximate zonal mean altitude value. In the revised manuscript, we have modified this sentence to “At each latitude and pressure level, we zonally average the altitude data (y-axis) from MCS”.

Figure 2: The caption refers to supplementary figure 4, but this ought to be SF 5. Change “The entire period starts from ... “ to “The period runs from Ls=195 to 252.” Actually, it is preferable to use Ls for the x-axis rather than the Martian day number.

Reply: Thanks for this suggestion. The sentence has been modified as suggested. And the x-axis has been modified to use Ls instead of Martian day number.

Figure 4: You might point out that the phases of U and V are consistent with those shown in supplementary figure 6. The figure would be simpler if the x-axis were simply labeled with local time. Of course, longitude and local time are equivalent for the migrating tide. The wind fields in panels c and d should be simply phase-shifted versions of each other, as they appear to be. I expect that the dust field at the two times 12 hours apart would also be phase shifted versions of each other, since possible vertical motion is not being accounted for.

Reply: Thanks for the suggestions and comments. We have pointed out in Fig. 4 that the phases of U and V are consistent with those shown in Supplementary Fig. 8 (corresponding to Supplementary Fig. 6 in the old manuscript). And we have made modifications to Fig. 4 to make it not so busy but retain the longitude annotation in the x-axis. We think it may be better to show the global map to the broad audiences and make it clearer that this is a global phenomenon rather than a local process. Merely using the local time as the x-axis may undermine this intension. Besides, it would be pointless to show two universal time (00:00 and 12:00) separately in two sub-figures if we just show the local time. Speaking of which, the wind fields in panels c and d are indeed simply phase-shifted version of each other but not the dust field. We have done tests on this but dust field at two local times with 12 hours apart could never be exact phase shifted versions of each other. The reason is that the simulation is run from a prescribed uniform dust distribution. The spin-up time for the simulation to generate a relatively stable westward propagating dust wave pattern is short. But it takes longer time to finally reach an ideal state that each dust particle can follow its own specific trajectory for every period. And since we can only derive the wind fields between 17.5°S to 72.5°S as shown in Fig. 4 because of the data availability and the method applicability, the wind circulation is not completed for the Lagrangian particle analysis to reach the final stable state. More specifically, we are unable to get it back when a dust particle goes southern than 72.5°S and this will cause a slow leakage of dust particles between 17.5°S to 72.5°S. And this is partly the reason why we choose a regional dust storm rather than a global dust storm to perform this Lagrangian particle analysis, because the meridional wind is much larger in a global dust storm and the dust particles are easier to go outside the available derived wind field zone. Therefore, we keep using the previous dust field results in Fig. 4c,d which has shown the wave-like pattern and the “simulated” dust front can match the observation well. We hope the reviewer is satisfied with this reply.

Supplementary Figure 2. The caption ought to spell out DH = Dust Height. Same for SF 4.

Reply: Modified as suggested.

Supplementary Figure 5. Remove “zoom-in” from the caption and simply state that this is the vertical and meridional structure of high latitude summer hemisphere dust opacity. I also suggest replacing “...during A regional dust storm...” with “...during the A-season regional dust storm...” It is preferable to use Ls for the x-axis rather than the Martian day number. This applies to Figure 2 as well.

Reply: Thanks for the suggestions. The caption has been rephrased. The “A regional dust storm” has been replaced by “the A-season regional dust storm” in all figures. The x-axis has been modified to use Ls instead of Martian day number.

Given the rather restricted local-time coverage, I believe it would be preferable to simply show the “am” and “pm” zonal mean dust fields, as is done in the Kleinbohl et al. [2019] abstract. Such a figure clearly captures the significant diurnal difference in the meridional and vertical extent of the high-latitude dust field.

Reply: Thanks for the suggestion. As discussed above in the reply to your comment for line 207, the data coverage in local time, longitude and altitude we used during the A regional dust storm of MY33 is better than that in the global dust storm of MY34. We have also added a new Supplementary Fig. 7 to show the data binning strategy for Supplementary Fig. 6 (the old Supplementary Fig. 5). We think this figure can give a better illustration on data availability and show where the uncertainty comes from and to what degree it can be. The potential uncertainties caused by the unevenly distributed data point within each bin grid is represented by the horizontal (equal to the local time bin interval) and vertical (according to the latitude bin interval) error bars for reference in the manuscript. We still think (if possible) it worthwhile to have a more detailed observation (more local times and higher temporal resolution than merely “am” and “pm”) on the diurnal variation of the dust front for comparison with simulations (as shown in Fig. 4c,d and Supplementary Fig. 10). Therefore, we retain the Supplementary Fig. 6 in the revised manuscript.

References:

- Banfield, D., B.J. Conrath, M.D. Smith, P.R. Christensen, and R.J. Wilson, 2003: Forced waves in the martian atmosphere from MGS TES nadir data, *Icarus*, 161, 319-345.
- Barnes, J. et al., 2017: The General Circulation. *The Atmosphere and Climate of Mars*.
- Kleinböhl, A., A. Spiga, D. M. Kass, J. H. Shirley, E. Millour, L. Montabone, and F. Forget (2019). Diurnal variations of dust from Mars Climate Sounder Observations during the global dust event in Mars Year 34. <https://www.hou.usra.edu/meetings/ninthmars2019/pdf/6146.pdf>
- Wilson, R.J., and M.I. Richardson, 2000: The Martian Atmosphere During the Viking Mission, 1: Infrared Measurements of Atmospheric Temperatures Revisited. *Icarus*, 145, 555-579.
- Wilson, R.J., R.M. Haberle, J. Noble, A.F.C. Bridger, J. Schaeffer, J.R. Barnes, and B.A. Cantor, Simulation of the 2001 planet-encircling dust storm with the NASA/NOAA Mars general circulation model, Mars atmosphere: modeling and observations workshop, Williamsburg, VA, November, 2008. <http://www.lpi.usra.edu/meetings/modeling2008/pdf/9023.pdf>
- Wilson, R.J. Martian dust storms, thermal tides and the Hadley circulation, Abstract 8069, Comparative Climatology of Terrestrial Planets, Boulder, CO, June 2012. <http://www.lpi.usra.edu/meetings/climatology2012/pdf/8069.pdf>

Reviewers' comments:

Reviewer #1 (Remarks to the Author):

In this manuscript, Wu et al. demonstrate that during major dust storms on Mars, a volume of high dust concentrations (or "dust front") migrates westward along with the Sun in line with the main westward-migrating diurnal tide. Mirroring this dust front, less dusty air migrates westward opposite to the Sun on the nightside. The conclusion drawn by the authors is that the dust front originates from rapid meridional transport at 10-40 km altitude (winds of almost 90 m/s) from dusty air in the mid-latitudes. In that case, less dusty and presumably moist air from the summer polar cap is likely being rapidly transported on the nightside to the mid-latitudes. Such a water supply, if entrained by dusty deep convection at low-mid latitudes, will effectively bring water from the summer polar cap to high in the middle atmosphere, where it can be photodissociated and enhance the supply of hydrogen that can escape from Mars's atmosphere.

The revised manuscript is a substantial improvement on the previously submitted version and reads a bit more smoothly, too. The revised manuscript makes three significant concessions to the concerns of the Reviewers. First, analyses based on indirect diagnosis of the water vapor distribution from the water ice opacity distribution are omitted, but the revised manuscript demonstrates still demonstrates the potential for water exchange based on archived global climate model output.

Second, the revised manuscript includes analyses of diurnal variability with a narrower time-averaging window. These analyses show that the diurnal variability they observe can be recovered from in-track observations alone. The in-track observations can be rigorously distinguished from the cross-track observations in MCS data by using the OBS_QUAL flag, but the time window approach used in the revised manuscript should be nearly equivalent.

Third, the revised manuscript includes analysis of output from appropriate Mars GCM simulations archived as the Mars Climate Database. This analysis shows good agreement with the results of the simplified modeling approaches otherwise used.

I therefore advise that this manuscript is suitable for publication in Nature Comms.

Let me briefly outline the significance of this result in the history of atmospheric dynamics. The exchange of air between the mid to high latitudes on Earth is strongly dependent on the behavior of mid-latitude cyclones and the jet stream along which they travel: a matter primarily of Rossby wave dynamics. While thermal tides were predicted for the Earth in the 18th century. It required careful barometry in the tropics to detect them.

Mars's thinner atmosphere certainly increases the power of the thermal tides, but until the last few years, the thermal tides were believed to be strongest in the tropics and mainly significant for the dynamics there. Extratropical Rossby wave activity is weaker in the summer than the winter and stronger in the northern hemisphere than the southern hemisphere. So one would not expect strong mid-high latitude exchange in the southern hemisphere during spring and summer. But apparently, thanks to the tides during dust storm activity, our intuition about what is important in a planetary atmosphere not our own is turned on its head.

Nicholas G. Heavens

Reviewer #2 (Remarks to the Author):

Re-review of manuscript "Dust Tides and Fast Meridional Motions in the Martian Atmosphere During Major Dust Storms" submitted to Nature Communications

The authors have revised their manuscript, taking several of my comments and the comments of other reviewers into account. The description of the averaging technique is improved by including the new figure 7 in the supplemental material. Rebinning and restricting the time range to one hour in figure 3 helps to compare apples to apples and reduce potential biases introduced due to the binning. Restricting the Ls range of some of the panels in figure 1 to avoid interpreting the onset of the dust storm would make sense, however, in the panels that were changed only 1-2 degrees of Ls were taken off. Large regional dust storms ramp up in a week or so such that the rationale for these small changes is not understandable.

The manuscript now contains comparisons with output from the Mars Climate Database. I would have envisioned that a GCM run is studied specifically for the dust storm in MY33 that is the focus of the manuscript, and these data would probably be available from the creators of the MCD. However, the MCD wind fields shown in supplementary figure 9 do agree surprisingly well with the winds derived from tidal theory, providing at least some validation for the approach. The questionable discussion of the vertical wind was removed from the manuscript.

The flawed analysis of water vapor derived from MCS cloud measurements was removed. It was replaced by a discussion of water vapor fields derived from the Mars Climate Database that argues that the diurnal tide would transport "moist air near the summer pole to be rapidly transported to lower latitudes during the night, where it then can be lifted by deep convection during the day and contribute to hydrogen escape from Mars during global dust storms." This claim is not supported by the data or the analysis presented in the manuscript. Supplementary figures 10a,b show that water around Ls=225 is higher at mid- and low latitudes than at polar latitudes. Supplementary figure 10c shows enhanced water in the polar region around Ls=280 but the tidal variation moves it only between 60S and 80S. Supplementary figure 10d suggests that the tidal effect of the global dust storm of Mars year 28 moves water all the way to the equator around Ls=280. However, the timing is not favorable for the mechanism suggested in the manuscript: The water reaches the lowest latitudes around 5 am and then starts moving again poleward. Water transport to high altitudes by the process of dusty deep convection would be expected to occur around noon to the early afternoon, when solar heat input is at maximum. By that time the low latitude water has decreased again by two thirds of the difference between minimum and maximum. In addition, tidal transport is predominantly cyclical, and requires additional dissipating processes to enable permanent transport. While the manuscript mentions this, it does not provide any analysis of such processes. However, this would be required in order to evaluate whether the tidal process has any significant role in water transport to the upper atmosphere and atmospheric escape. The way it is described in the manuscript it is not more than a hypothesis.

Finally, I do want to point out that the manuscript on this topic that was mentioned by reviewer 1 is now published. It is available on the website of the Journal of Geophysical Research as an accepted article with the doi:10.1029/2019JE006115 and a publication date of October 18, 2019. It discusses strong diurnal variations in the vertical and latitudinal distribution of dust during the 2018 global dust storm and evaluates diurnal tidal variations as the origin of the observed behavior in the south polar region using a General Circulation Model. The publication of the JGR article reduces the novelty of the effect discussed in this manuscript significantly. The current manuscript mainly shows that the processes discussed in the JGR article for the 2018 global storm also apply, with a lower magnitude, to various regional dust storms. In light of this, and the lack of a convincing claim concerning the water vapor transport to the middle atmosphere, I do not consider this manuscript to be suitable for publication in a Nature journal.

Reviewer #3 (Remarks to the Author):

Dust Tides and Fast Meridional Motions in the Martian Atmosphere During Major Dust Storms,
Revised by Zhaopeng Wu, Tao Li, Xi Zhang and Jun Cui.

The authors have made a good effort to respond to the reviews and have made significant improvements to the original submission. However I feel that the paper would benefit from another round of revision and I encourage consideration of the following.

I appreciated the additional use of Mars global circulation model results to buttress the idealized tide modeling, in the midlatitude region of relevance. In particular, the comparison in Supplementary Figure 9 is reassuring.

P. 21: It is stated that the meridional wind is “dominated only by the reconstructed DW1 meridional winds (Fig. 4b)...”. Of course, the referenced figure only shows the tidal component of the V field, however it is not clearly stated that the diurnal . It is obvious (and not surprising) that Supplementary Figure 9 shows that the ideally approximated V field in the tropics (equatorward of 30 degrees) differs significantly from that in the MCD GCM simulation. I accept that advection by the tropical tide winds is not within the scope of the paper.

I agree with the authors that the submitted paper treats the maintenance of a zonally varying dust front in more generality than previous work and is thus worthy of publication. However I do feel that it is appropriate to comment on and reference the conference abstract by Kleinbohl et al. [2019], which includes a figure very similar to Supplementary figure 6. I’d further note that reference 16 (also a conference abstract, but from 2009), noted in passing in the first paragraph of page 3 shows a very similar figure based on MGS TES limb observations of temperature and dust. That work used also GCM modeling to suggest the likely role of the thermal tide in modulating the high latitude dust distribution.

The data processing (specifically temporal binning) is appropriate and its presentation has been improved. As was noted in the author’s comments to the reviewers, the binning strategy was devised to demonstrate that the migrating diurnal tide is the dominant component of zonal structure in the temperature field, and, by extension, the wind field. This ought to be more explicitly stated in the manuscript. I had suggested the authors present some basic discussion about tides for a more broad audience. I don’t thin that they have been very successful. A critical point is that the diurnal migrating tide (which is perhaps more intuitively known also as the diurnal period sun-synchronous tide) directly responds to thermal forcing by the zonally-averaged component of aerosol, hence its close dependence on dust storm activity. A key point in its identification is isolating the zonally-averaged diurnal temperature in a reference frame with observations at fixed local solar times. The main thrust of the data analysis should be to stress that under dust storm conditions, the vertically trapped component of the migrating temperature tide (present at mid and high latitudes, but not the tropics), is dominant and readily isolated in the observations. Tide theory then usefully allows the associated horizontal velocity fields to be synthesized.

Minor Comments:

P.17: It is noted that wind observations are effectively non-existent. Therefore it is unnecessary to state that it is unrealistic to obtain diurnally-varying wind from observations

P. 10: Discussion section: Concerning the derived wind fields based on “our simple tidal theories”. The relationship between diurnally-varying winds and temperature is well established in classical tide theory. There is nothing new here...other than the significant demonstration that the theory provides useful guidance for the velocity fields in the midlatitudes. This latter point might be more strongly emphasized.

P. 11: Meridional advection of dust, water vapor and wind is expected on a diurnal time scale. The net transport of these fields would need to be accomplished by circulator elements other than the

regularly oscillating thermal tide. I believe that is what the authors are getting at in the lower part of the page. This would require much more sophisticated modeling, such as with a global circulation model.

P. 11: What is meant by "...strong circulations asymmetric with the westward-propagating diurnal tide ..."? The sentence that follows is not a fully formed expression.

Reference 16: Should be expanded to include the title of the paper, which is particularly relevant. McConnochie, T.H., Wilson, R.J. & Smith, M.D (2009) Dust in the MGS-TES limb sounding data set: Dust advection by thermal tides and the dust-free winter pole, in Final Conference Proceeding of the Mars Dust Cycle Workshop, NASA/Ames Research Center, CA. 152-155.

https://spacescience.arc.nasa.gov/mars-climate-modeling-group/documents/mars_dust_cycle_workshop_abstracts.pdf

Reference 40 should be : Mars atmosphere: modeling and observations workshop, Williamsburg, VA, November, 2008. <http://www.lpi.usra.edu/meetings/modeling2008/pdf/9023.pdf>Not Journal of Applied Microbiology!

Responses to reviewers (in blue)

Reviewers' comments:

Reviewer #1 (Remarks to the Author):

In this manuscript, Wu et al. demonstrate that during major dust storms on Mars, a volume of high dust concentrations (or "dust front") migrates westward along with the Sun in line with the main westward-migrating diurnal tide. Mirroring this dust front, less dusty air migrates westward opposite to the Sun on the nightside. The conclusion drawn by the authors is that the dust front originates from rapid meridional transport at 10-40 km altitude (winds of almost 90 m/s) from dusty air in the mid-latitudes. In that case, less dusty and presumably moist air from the summer polar cap is likely being rapidly transported on the nightside to the mid-latitudes. Such a water supply, if entrained by dusty deep convection at low-mid latitudes, will effectively bring water from the summer polar cap to high in the middle atmosphere, where it can be photodissociated and enhance the supply of hydrogen that can escape from Mars's atmosphere.

The revised manuscript is a substantial improvement on the previously submitted version and reads a bit more smoothly, too. The revised manuscript makes three significant concessions to the concerns of the Reviewers. First, analyses based on indirect diagnosis of the water vapor distribution from the water ice opacity distribution are omitted, but the revised manuscript demonstrate still demonstrates the potential for water exchange based

on archived global climate model output.

Second, the revised manuscript includes analyses of diurnal variability with a narrower time-averaging window. These analyses show that the diurnal variability they observe can be recovered from in-track observations alone. The in-track observations can be rigorously distinguished from the cross-track observations in MCS data by using the OBS_QUAL flag, but the time window approach used in the revised manuscript should be nearly equivalent.

Third, the revised manuscript includes analysis of output from appropriate Mars GCM simulations archived as the Mars Climate Database. This analysis shows good agreement with the results of the simplified modeling approaches otherwise used.

I therefore advise that this manuscript is suitable for publication in Nature Comms.

Let me briefly outline the significance of this result in the history of atmospheric dynamics.

The exchange of air between the mid to high latitudes on Earth is strongly dependent on the behavior of mid-latitude cyclones and the jet stream along which they travel: a matter primarily of Rossby wave dynamics. While thermal tides were predicted for the Earth in the 18th century. It required careful barometry in the tropics to detect them.

Mars's thinner atmosphere certainly increases the power of the thermal tides, but until the

last few years, the thermal tides were believed to be strongest in the tropics and mainly significant for the dynamics there. Extratropical Rossby wave activity is weaker in the summer than the winter and stronger in the northern hemisphere than the southern hemisphere. So one would not expect strong mid-high latitude exchange in the southern hemisphere during spring and summer. But apparently, thanks to the tides during dust storm activity, our intuition about what is important in a planetary atmosphere not our own is turned on its head.

Nicholas G. Heavens

We greatly thank Dr. Nicholas Heavens for his constructive suggestions and insightful comments which helped us improve the manuscript substantially. The comments outlined above that include an historical view of this air exchange problem can improve the significance of our results to a higher level. We have incorporated them in the discussion section since it gives this paper a strong ending to broad audiences.

Reviewer #2 (Remarks to the Author):

Re-review of manuscript “Dust Tides and Fast Meridional Motions in the Martian Atmosphere During Major Dust Storms” submitted to Nature Communications

The authors have revised their manuscript, taking several of my comments and the comments of other reviewers into account. The description of the averaging technique is improved by including the new figure 7 in the supplemental material. Rebinning and restricting the time range to one hour in figure 3 helps to compare apples to apples and reduce potential biases introduced due to the binning. Restricting the Ls range of some of the panels in figure 1 to avoid interpreting the onset of the dust storm would make sense, however, in the panels that were changed only 1-2 degrees of Ls were taken off. Large regional dust storms ramp up in a week or so such that the rationale for these small changes is not understandable.

Reply: Thanks for this comment. We believe that figure 1c,d,e are these panels mentioned by the reviewer that were changed only 1-2 degrees of Ls. In the revised version, we have made stronger restrictions to the Ls range of these panels by cutting off more days which are probably in the “onset” periods. Considering the availability of the cross-track observation periods (Supplementary Fig. 4) and relatively short duration time for some of these regional dust storms (Supplementary Fig. 3), we also change the examples of 10 days of diurnal change in dust opacity to examples of 8 days both in Figure 1 and Supplementary Figure 1. Then the change of Ls in figure 1c,d,e from that of the first round’s review are up to 3 degrees, which should be more understandable. The change from 10 days’ example to 8 days’ example in Figure 1 and Supplementary Figure 1 does not change any conclusion of the manuscript but makes the figures more concise and the diurnal variation of dust opacity more prominent.

The manuscript now contains comparisons with output from the Mars Climate Database. I

would have envisioned that a GCM run is studied specifically for the dust storm in MY33 that is the focus of the manuscript, and these data would probably be available from the creators of the MCD. However, the MCD wind fields shown in supplementary figure 9 do agree surprisingly well with the winds derived from tidal theory, providing at least some validation for the approach. The questionable discussion of the vertical wind was removed from the manuscript.

The flawed analysis of water vapor derived from MCS cloud measurements was removed. It was replaced by a discussion of water vapor fields derived from the Mars Climate Database that argues that the diurnal tide would transport “moist air near the summer pole to be rapidly transported to lower latitudes during the night, where it then can be lifted by deep convection during the day and contribute to hydrogen escape from Mars during global dust storms.” This claim is not supported by the data or the analysis presented in the manuscript. Supplementary figures 10a,b show that water around $L_s=225$ is higher at mid- and low latitudes than at polar latitudes. Supplementary figure 10c shows enhanced water in the polar region around $L_s=280$ but the tidal variation moves it only between 60S and 80S. Supplementary figure 10d suggests that the tidal effect of the global dust storm of Mars year 28 moves water all the way to the equator around $L_s=280$. However, the timing is not favorable for the mechanism suggested in the manuscript: The water reaches the lowest latitudes around 5 am and then starts moving again poleward. Water transport to high altitudes by the process of dusty deep convection would be expected to occur around noon to the early afternoon, when solar heat input is at maximum. By that time the

low latitude water has decreased again by two thirds of the difference between minimum and maximum. In addition, tidal transport is predominantly cyclical, and requires additional dissipating processes to enable permanent transport. While the manuscript mentions this, it does not provide any analysis of such processes. However, this would be required in order to evaluate whether the tidal process has any significant role in water transport to the upper atmosphere and atmospheric escape. The way it is described in the manuscript it is not more than a hypothesis.

Reply: Thanks for this comment. We agree to the reviewer that in the period around noon to the early afternoon when dusty deep convection would be expected to occur mostly, the water vapor is in the retreat phase to higher latitudes from the equatorial region as shown in Supplementary Figure 10d. However, we want to point out that there are still large amount of water vapor at low-to-mid latitudes between 20°S-60°S by that time (compare the meridional distributions of water vapor at two longitude -180°[noon] and -90°[18LT] in Supplementary Figure 10d). More apparently, by comparison between Supplementary figures 10 d and c, we can see that the diurnal meridional wind in no dust storm condition (c) is weak and unable to reach its hand to water vapor reservoir at high latitudes (greater than 60°S), while the enhanced diurnal wind circulation during global dust storms (d) is strong enough to transport the water vapor from the high latitude reservoir and widely spread it to almost the whole Southern Hemisphere. In addition, as the reviewer mentioned that the dusty deep convection was expected to occur when solar heat input is at maximum, we want to point out that around southern summer solstice ($L_s=270^\circ$), the subsolar point is at $\sim 25^\circ\text{S}$ on Mars, which means the solar heat input maximum is near the border between the low and middle latitudes and contribute to the occurrence of dusty deep convection in these regions. According to recent observational researches [e.g., Heavens et al., 2019], the dusty deep convection becomes more widespread during global and regional dust storms at southern midlatitudes. In addition, modeling work [Wang et al., 2018, Figure 3] also suggested that deep convections can occur at midlatitudes during summer time. Therefore, the water vapor in the southern middle latitudes is also susceptible to deep convections.

Back to the issue of latitude range of water transport around noon, we show below the “Figure 1 for the reviewer” (hereafter abbreviated as RFigure 1) to illustrate the complexity and diversity in this issue. Rfigure 1a is at the same time as Supplementary Figure 10d but in a different altitude (20Pa versus 50 Pa). The water horizontal distribution changes as the wind field becomes stronger at 20 Pa than that at 50 Pa. The nightside tidal wind can move the water northward further, even to the Northern Hemisphere. We can see there is still large amount of water remained in the low latitudes around noon to the early afternoon (between -180° and -150° longitude in RFigure 1a).

Figure 1b is at the same altitude as Supplementary Figure 10d but at slightly earlier L_s ($L_s=270^\circ$ versus $L_s=280^\circ$). We can see the water distribution also changed a bit. Most of all, there's a water maximum around noon to the early afternoon at $30^\circ\text{S} - 40^\circ\text{S}$ latitudes, not far from the sub-solar point, which means this water maximum is highly susceptible to deep convections. Figure 1c,d show cases of another global dust storm in MY25. It should be noted that the MY25 global dust storm mainly occurred in southern spring, not in southern summer. However, we can also find the similar day-night transport of water vapor in the ending period of this global dust storm. The chosen L_s in Figure 1c,d is near the southern summer solstice so water vapor concentration in the southern high latitudes has increased and the process that nightside tidal wind moves water to lower latitudes is valid as well. This further prove that the mechanism that we proposed is common on Mars.

Figure 1 for the reviewer | Diurnal variation of the horizontal wind fields (blue vectors) and water vapor distribution (colors) during global dust storms in MY28 (a, b) and MY25 (c, d) from MCD 5.3. **a**, 20 Pa, 00:00 UT, $L_s=280^\circ$ in MY28. **b**, 50 Pa, 00:00 UT, $L_s=270^\circ$ in MY28. **c**, 50 Pa, 00:00 UT, $L_s=250^\circ$ in MY25. **d**, 50 Pa, 00:00 UT, $L_s=260^\circ$ in MY25. The blank in the polar region in subfigure **d** indicates higher water vapor content that exceed the range of color bar.

As a conclusion, the circumstances of the water vapor distribution and evolution within a sol (specifically, the latitude range it can reach around noon to early afternoon) differ in different L_s and MYs and even at different altitudes. And the influence of dusty deep convections on the vertical transport of water vapor in different latitudes and seasons is also complicated. The mechanism should be more complicated in reality and for different dust storms thus need a lot of future works. The exact amount of water vapor into this process is hard to evaluated for now for lack of observations of diurnal change of global water vapor. As for the “additional dissipating

processes to enable permanent transport”, it is true we do not provide the detailed analysis on this process. A comprehensive and high-resolution GCM simulations with all those vertical processes including deep convections properly parameterized in the model is necessary to dragonize this issue qualitatively and quantitatively, and this is far beyond the scope of this work. Our intention here is to address this hypothesis and motivate a potential direction for research on the short time-scale atmospheric dynamics.

However, we thank the reviewer for pointing out the specific mistakes, e.g., “Supplementary figure 10c shows enhanced water in the polar region around $L_s=280$ but the tidal variation moves it only between 60S and 80S.” We should have described the Supplementary figure 10c more clearly in the previous version and pointed out that the nightside tidal wind can only transport water vapor from the higher to lower latitudes in a limited range under no-dust-storm condition. This has been corrected in the revised version. In addition, we have added some part of the reply to the reviewer above to the revised manuscript to make the discussions on water vapor transport clearer and sound.

Reference:

Heavens, N. G., Kass, D. M., Shirley, J. H., Piqueux, S. & Cantor, B. A. An Observational Overview of Dusty Deep Convection in Martian Dust Storms. *J Atmos Sci* (2019)

Wang, C. et al. Parameterization of Rocket Dust Storms on Mars in the LMD Martian GCM: Modeling Details and Validation. *Journal of Geophysical Research: Planets* 123, 982-1000, doi:10.1002/2017je005255 (2018).

Finally, I do want to point out that the manuscript on this topic that was mentioned by reviewer 1 is now published. It is available on the website of the *Journal of Geophysical Research* as an accepted article with the doi:10.1029/2019JE006115 and a publication date of October 18, 2019. It discusses strong diurnal variations in the vertical and latitudinal distribution of dust during the 2018 global dust storm and evaluates diurnal tidal variations as the origin of the observed behavior in the south polar region using a General Circulation Model. The publication of the JGR article reduces the novelty of the effect discussed in this manuscript significantly. The current manuscript mainly shows that the processes discussed in the JGR article for the 2018 global storm also apply, with a lower

magnitude, to various regional dust storms. In light of this, and the lack of a convincing claim concerning the water vapor transport to the middle atmosphere, I do not consider this manuscript to be suitable for publication in a Nature journal.

Reply: Thanks for the comment.

First, as has been told by the editor, we do not need to concern about the novelty issue because of the scoop policy Nature Communications operates. However, as suggested by Reviewer #3 and the editor, we have cited and commented the relevant study [Kleinböhl et al., 2019] in our revised manuscript.

Second, we want to emphasize again the significance of our work for broad audiences:

(a) We discovered and discussed rapidly migrating dust fronts in all major dust storms in total 7 Mars Years, which makes this phenomenon more general and significant.

(b) Instead of using comprehensive GCM simulation [McConnochie et al., 2009; Kleinböhl et al., 2019], we use observational data and readily comprehensible classical theories to provide a direct demonstration on the relationship between the diurnal thermal tide and the diurnal variation of dust front (Figure 2 and 4 in the main text). Based on these evidences we can describe the phenomenon as “dust tide”, which provides a more intuitive illustration to broad audiences.

(c) We extended the dust phenomenon to a broader scope by proposing a potentially rapid mechanism for water vapor escape, which is very important for Mars climate evolution. This hypothesis may help motivate a potential direction for research on the short time-scale atmospheric dynamics.

In light of these, we hope the current manuscript is acceptable for publication.

Reference:

McConnochie, T.H., Wilson, R.J. & Smith, M.D (2009) Dust in the MGS-TES limb sounding data set: Dust advection by thermal tides and the dust-free winter pole, in Final Conference Proceeding of the Mars Dust Cycle Workshop, NASA/Ames Research Center, CA. 152-155.
https://spacescience.arc.nasa.gov/mars-climate-modeling-group/documents/mars_dust_cycle_workshop_abstracts.pdf

Kleinböhl, A. et al. Diurnal Variations of Dust during the 2018 Global Dust Storm observed by the Mars Climate Sounder. *Journal of Geophysical Research: Planets* n/a, doi:10.1029/2019je006115.

Reviewer #3 (Remarks to the Author):

Dust Tides and Fast Meridional Motions in the Martian Atmosphere During Major Dust Storms, Revised by Zhaopeng Wu, Tao Li, Xi Zhang and Jun Cui.

The authors have made a good effort to respond to the reviews and have made significant improvements to the original submission. However I feel that the paper would benefit from another round of revision and I encourage consideration of the following.

I appreciated the additional use of Mars global circulation model results to buttress the idealized tide modeling, in the midlatitude region of relevance. In particular, the comparison in Supplementary Figure 9 is reassuring.

P. 21: It is stated that the meridional wind is “dominated only by the reconstructed DW1 meridional winds (Fig. 4b)...”. Of course, the referenced figure only shows the tidal component of the V field, however it is not clearly stated that the diurnal . It is obvious (and not surprising) that Supplementary Figure 9 shows that the ideally approximated V field in the tropics (equatorward of 30 degrees) differs significantly from that in the MCD GCM simulation. I accept that advection by the tropical tide winds is not within the scope of the paper.

Reply: Thanks for this comment. We have changed “the meridional wind field is dominated only by the reconstructed DW1 meridional winds (Fig. 4b)” to “the meridional wind field is approximate to the reconstructed DW1 meridional winds (Fig. 4b)”. And we added in this section some necessary discussion: “Note that from the discussion on the potential biases of wind results

in the tropics in the above section and Supplementary Fig. 9, the derived DW1 meridional wind field in the mid-to-high latitudes is validated by the MCD simulation, while in the tropics (equatorward of 30°), it differs significantly from that of the MCD. Therefore, the results based on the derived wind field in the tropics should be treated with caution.”

I agree with the authors that the submitted paper treats the maintenance of a zonally varying dust front in more generality than previous work and is thus worthy of publication. However I do feel that it is appropriate to comment on and reference the conference abstract by Kleinbohl et al. [2019], which includes a figure very similar to Supplementary figure 6. I'd further note that reference 16 (also a conference abstract, but from 2009), noted in passing in the first paragraph of page 3 shows a very similar figure based on MGS TES limb observations of temperature and dust. That work used also GCM modeling to suggest the likely role of the thermal tide in modulating the high latitude dust distribution.

Reply: Thanks for the valuable suggestions. We have added the comment and citation of Kleinbohl et al. [2019] in the discussions of Figure 3 and Supplementary figure 6, and the citation of reference 16 to the discussion of Figure 3. And we also introduced these two works in the introduction section, e.g. “Numerical simulation studies also suggested the possible role of the thermal tide in modulating the mid-to-high-latitude dust distribution^{15,16}”

The data processing (specifically temporal binning) is appropriate and its presentation has been improved. As was noted in the author's comments to the reviewers, the binning strategy was devised to demonstrate that the migrating diurnal tide is the dominant component of zonal structure in the temperature field, and, by extension, the wind field. This ought to be more explicitly stated in the manuscript. I had suggested the authors present some basic discussion about tides for a more broad audience. I don't think that they have been very successful. A critical point is that the diurnal migrating tide (which is

perhaps more intuitively known also as the diurnal period sun- synchronous tide) directly responds to thermal forcing by the zonally-averaged component of aerosol, hence its close dependence on dust storm activity. A key point in its identification is isolating the zonally-averaged diurnal temperature in a reference frame with observations at fixed local solar times. The main thrust of the data analysis should be to stress that under dust storm conditions, the vertically trapped component of the migrating temperature tide (present at mid and high latitudes, but not the tropics), is dominant and readily isolated in the observations. Tide theory then usefully allows the associated horizontal velocity fields to be synthesized.

Reply: Thanks for these useful comments and suggestions. We have added them to the revised manuscript (mainly in the introduction section, highlighted in red).

Minor Comments:

P.17: It is noted that wind observations are effectively non-existent. Therefore it is unnecessary to state that it is unrealistic to obtain diurnally-varying wind from observations

Reply: Thanks for this suggestion. We have deleted this unnecessary sentence “It’s unrealistic to directly obtained the daily varied winds from observations” in the revised manuscript.

P. 10: Discussion section: Concerning the derived wind fields based on “our simple tidal theories”. The relationship between diurnally-varying winds and temperature is well established in classical tide theory. There is nothing new here...other than the significant demonstration that the theory provides useful guidance for the velocity fields in the

midlatitudes. This latter point might be more strongly emphasized.

Reply: Thanks for pointing out this careless mistake. In the revised version, we have replaced “our simple tidal theories” with “the classical theories” and modified this sentence to “The good agreement between the derived wind fields based on the classical theories and the numerical simulations from MCD confirms that (1) the classical gradient and tidal theories are valid for qualitatively analysing the dust tides in the mid-latitudes on Mars...”

P. 11: Meridional advection of dust, water vapor and wind is expected on a diurnal time scale. The net transport of these fields would need to be accomplished by circulation elements other than the regularly oscillating thermal tide. I believe that is what the authors are getting at in the lower part of the page. This would require much more sophisticated modeling, such as with a global circulation model.

P. 11: What is meant by “...strong circulations asymmetric with the westward-propagating diurnal tide ...”? The sentence that follows is not a fully formed expression.

Reply: Thanks for the comments. We have rewritten the corresponding sentences according to the suggestions of the reviewer in the revised discussion section (highlighted in red).

Reference 16: Should be expanded to include the title of the paper, which is particularly relevant. McConnochie, T.H., Wilson, R.J. & Smith, M.D (2009) Dust in the MGS-TES limb sounding data set: Dust advection by thermal tides and the dust-free winter pole, in Final Conference Proceeding of the Mars Dust Cycle Workshop, NASA/Ames Research Center, CA. 152-155. https://spacescience.arc.nasa.gov/mars-climate-modeling-group/documents/mars_dust_cycle_workshop_abstracts.pdf

Reference 40 should be : Mars atmosphere: modeling and observations workshop, Williamsburg, VA, November, 2008. <http://www.lpi.usra.edu/meetings/modeling2008/pdf/9023.pdf>Not Journal of Applied

Microbiology!

Reply: Thanks for pointing the mistakes out. We have modified them in the revised manuscript.

REVIEWERS' COMMENTS:

Reviewer #2 (Remarks to the Author):

Re-review of manuscript "Dust Tides and Fast Meridional Motions in the Martian Atmosphere During Major Dust Storms" submitted to Nature Communications

I have re-reviewed the revised manuscript. However, I do not think my main point of concern, the transport of water vapor from the polar regions to the low latitudes where it would be available for upward transport in deep dusty convection, has been addressed adequately. While some of the points I had raised concerning this issue have been included into the extended discussion section of the revised manuscript (lines 242-262), the discussion mainly qualifies statements made previously, rather than providing convincing arguments for this process to be relevant. In southern spring water vapor is located at low latitudes no matter whether a dust storm is occurring or not (e.g. supplementary figures 10 a,b) so the process is not really important in this season. In southern summer the water vapor is more concentrated at southern polar latitudes as shown in their supplementary figure 10 c. This pattern is obviously modified by the dust storm in MY 28 (supplementary figure 10 d) but the timing is not favorable. The manuscript states that "the dusty deep convection becomes more widespread during global and regional dust storms at southern midlatitudes" and Heavens et al. (2019) is provided as a reference. While this is not incorrect, it is also true that deep dusty convection increases over a wide latitude range that includes the northern midlatitudes, as evidenced for MY 29 in figure 7 of the aforementioned manuscript and for the MY 34 global dust storm as shown in a recent JGR manuscript by the same author. So I am still not convinced about the relevance of the process for actual water transport in dusty deep convection that is put forward by the manuscript.

The authors have created a new figure that is included in their review response, in which they provide data excerpts from the Mars Climate Database at higher altitudes (20 Pa) as well as for the MY 25 global dust storm. I appreciate the effort. However, while the data for MY 28 shows slightly more water vapor at 20 Pa in the early afternoon than at 50 Pa, it is questionable how systematic this behavior is. The MY 25 data does not show significant increases in water vapor mixing ratio north of 30 S between noon and midnight at either pressure level so I don't see how it helps the argument. The manuscript still does not provide a convincing argument for the claim that these tidal processes are significant for water vapor transport into the middle atmosphere by dusty deep convection. Without this connection solidly established, the work is largely an extension and generalization of the work published on the MY 34 global dust storm, which I would consider interesting for the specialist but not significant for broad audiences. Taking these two findings together I do not consider this manuscript to be suitable for publication in a Nature journal.

Reviewer #3 (Remarks to the Author):

I am satisfied with the current revision.

Responses to reviewers (in blue)

REVIEWERS' COMMENTS:

Reviewer #2 (Remarks to the Author):

Re-review of manuscript “Dust Tides and Fast Meridional Motions in the Martian Atmosphere During Major Dust Storms” submitted to Nature Communications

I have re-reviewed the revised manuscript. However, I do not think my main point of concern, the transport of water vapor from the polar regions to the low latitudes where it would be available for upward transport in deep dusty convection, has been addressed adequately. While some of the points I had raised concerning this issue have been included into the extended discussion section of the revised manuscript (lines 242-262), the discussion mainly qualifies statements made previously, rather than providing convincing arguments for this process to be relevant. In southern spring water vapor is located at low latitudes no matter whether a dust storm is occurring or not (e.g. supplementary figures 10 a,b) so the process is not really important in this season. In southern summer the water vapor is more concentrated at southern polar latitudes as shown in their supplementary figure 10 c. This pattern is obviously modified by the dust storm in MY 28 (supplementary figure 10 d) but the timing is not favorable. The manuscript states that “the dusty deep convection becomes more widespread during global and regional dust storms at southern midlatitudes” and Heavens et al. (2019) is

provided as a reference. While this is not incorrect, it is also true that deep dusty convection increases over a wide latitude range that includes the northern midlatitudes, as evidenced for MY 29 in figure 7 of the aforementioned manuscript and for the MY 34 global dust storm as shown in a recent JGR manuscript by the same author. So I am still not convinced about the relevance of the process for actual water transport in dusty deep convection that is put forward by the manuscript.

Reply: Thanks for these comments.

As for “In southern spring water vapor is located at low latitudes no matter whether a dust storm is occurring or not (e.g. supplementary figures 10 a,b) so the process is not really important in this season”, we agree with the reviewer. We have mentioned it in the discussion section of the previously revised manuscript that “the global distribution of water vapor shows clear seasonal variabilities, implying that the effect of meridional motion on water vapor might be different in different seasons” and “this may contribute to the rapid enhancement of water content at high altitudes and hydrogen escape **during the southern-summer-season global dust storms**, such as that observed in MY 28”. These discussions have specified that this process is only important during the southern summer season, when water vapor is mainly located at southern polar region. We also want to point out that due to significantly increased sublimation of polar cap water ice, the water vapor concentration in the southern-summer polar region is nearly 1 order of magnitude higher than that at southern-spring low latitudes. This makes the tidal transport of water vapor from the polar reservoir to lower latitudes during southern summer more important.

As for “This pattern is obviously modified by the dust storm in MY 28 (supplementary figure 10 d) but the timing is not favorable”, we disagree with the reviewer. In the period around noon to the early afternoon when dusty deep convection occurs more frequently, the water vapor is indeed in the retreat phase from low to high latitudes as shown in Supplementary Figure 10d. However, a large amount of water vapor is still located at low-to-mid latitudes between 20°S-60°S during that period (comparing the meridional distributions of water vapor at two longitude -180°[noon] and -90°[18LT] in Supplementary Figure 10d). Therefore, the widespread deep convection at southern midlatitudes is still able to lift the water vapor to high altitudes.

As for “The manuscript states that “the dusty deep convection becomes more widespread during global and regional dust storms at southern midlatitudes” and Heavens et al. (2019) is provided as a reference. While this is not incorrect, it is also true that deep dusty convection increases over a wide latitude range that includes the northern midlatitudes, as evidenced for MY 29 in figure 7 of

the aforementioned manuscript and for the MY 34 global dust storm as shown in a recent JGR manuscript by the same author”, we agree with the reviewer that the deep dusty convection increases not only at southern midlatitudes, but also at northern midlatitudes sometimes. However, this does not weaken our statement that the water vapor spread from southern high to low latitudes by the enhanced tidal wind is susceptible to deep convection at southern mid-to-low latitudes. Furthermore, the deep convection at northern mid-latitudes in MY 29 in figure 7 of Heavens et al. (2019) actually occurred at $L_s=145.32$, which is outside the dust storm season ($L_s=180-360$), and thus it is not the concern of our work.

As for “the discussion mainly qualifies statements made previously, rather than providing convincing arguments for this process to be relevant”, we believe our data analysis has qualitatively shown evidence that the meridional transport of water vapor is relevant to the upward water transport in deep dusty convection. But we also agree that more direct observations of diurnal change of global water vapor are needed in the future to quantify this process in detail.

The authors have created a new figure that is included in their review response, in which they provide data excerpts from the Mars Climate Database at higher altitudes (20 Pa) as well as for the MY 25 global dust storm. I appreciate the effort. However, while the data for MY 28 shows slightly more water vapor at 20 Pa in the early afternoon than at 50 Pa, it is questionable how systematic this behavior is. The MY 25 data does not show significant increases in water vapor mixing ratio north of 30 S between noon and midnight at either pressure level so I don't see how it helps the argument. The manuscript still does not provide a convincing argument for the claim that these tidal processes are significant for water vapor transport into the middle atmosphere by dusty deep convection. Without this connection solidly established, the work is largely an extension and generalization of the work published on the MY 34 global dust storm, which I would consider interesting for the specialist but not significant for broad audiences. Taking these two findings together I do not consider this manuscript to be suitable for publication in a Nature journal.

Reply: The new figure in the previous review response was provided to show the complexity and generalization of the proposed water transport mechanism. The different water vapor distribution at 20 Pa indicated the complexity in the vertical, while the data in MY 25 implied more generalization of proposed mechanism. We can see that even for the MY 25 global dust storm, the water vapor shows similar day-night distribution as that of MY 28. As has been discussed in the above responses, the dusty deep convection becomes much more widespread during global and regional dust storms at southern midlatitudes (30°S-60°S) [Heavens et al. 2019], not only at lower latitudes (north of 30°S). We can also see that a large amount of water vapor is still located at low-to-mid latitudes between 20°S-60°S during noon to the early afternoon and is highly susceptible to be lifted by deep convection, just like that in MY28. So our proposed water transport mechanism can be also applied to more major dust storms other than MY 28. As for the main text of our manuscript, the discussion section has provided qualitative evidence that the water vapor transported by nightside tidal wind from southern polar region to the mid-to-low latitudes is susceptible to daytime deep convection at southern mid-to-low latitudes during major dust storms. Both the short-term dynamical process on Mars including water transport and the use of simple classical theory to gain an intuitive understanding of dust tides should be interesting enough for broad audiences.